



# Improved monitoring of shipping NO$_2$ with TROPOMI: decreasing NO$_x$ emissions in European seas during the COVID-19 pandemic

T. Christoph V.W. Riess[1], K. Folkert Boersma[1,2], Jasper van Vliet[3], Wouter Peters[1,4], Maarten Sneep[2], Henk Eskes[2], and Jos van Geffen[2]

[1]Department of Meteorology and Air Quality, Wageningen University, Wageningen, the Netherland
[2]Climate Observations Department, Royal Netherlands Meteorological Institute, De Bilt, the Netherlands
[3]Human Environment and Transport Inspectorate, the Netherlands
[4]University of Groningen, Centre for Isotope Research, Groningen, the Netherlands

**Correspondence:** TCVW Riess (christoph.riess@wur.nl)

**Abstract.** TROPOMI measurements of tropospheric NO$_2$ columns provide powerful information on emissions of air pollution by ships on open sea. This information is potentially useful for authorities to help determine the (non-)compliance of ships with increasingly stringent NO$_x$ emission regulations. We find that the information quality is improved further by recent upgrades in the TROPOMI cloud retrieval and an optimal data selection. We show that the superior spatial resolution of TROPOMI allows

the detection of several lanes of NO$_2$ pollution ranging from the Aegean Sea near Greece to the Skagerrak in Scandinavia, which have not been detected with other satellite instruments before. Additionally, we demonstrate that under conditions of sun glint TROPOMI's vertical sensitivity to NO$_2$ in the marine boundary layer increases by up to 60%. The benefits of sun glint are most prominent under clear-sky situations when sea surface winds are low, but slightly above zero ($\pm 2$ m/s). Beyond spatial resolution and sun glint, we examine for the first time the impact of the recently improved cloud algorithm on the TROPOMI

NO$_2$ retrieval quality, both over sea and over land. We find that the new FRESCO+wide algorithm leads to 50 hPa lower cloud pressures, correcting a known high bias, and produces 1-4·10$^{15}$ molec·cm$^{-2}$ higher retrieved NO$_2$ columns, thereby at least partially correcting for the previously reported low bias in the TROPOMI NO$_2$ product. By training an artificial neural network on the 4 available periods with standard and FRESCO+wide test-retrievals, we develop a historic, consistent TROPOMI NO$_2$ data set spanning the years 2019 and 2020. This improved data set shows stronger (35-75%) and sharper (10-35%) shipping

NO$_2$ signals compared to co-sampled measurements from OMI. We apply our improved data set to investigate the impact of the COVID-19 pandemic on ship NO$_2$ pollution over European seas and find indications that NO$_x$ emissions from ships reduced by 20-25% during the pandemic. The reductions in ship NO$_2$ pollution start in March-April 2020, in line with changes in shipping activity inferred from AIS data.

**Keywords.** NO$_2$, TROPOMI, shipping, MARPOL Annex VI, sun glint, cloud properties, COVID-19

# 1 Introduction

Emissions of nitrogen oxides (NO$_x$=NO+NO$_2$) have several primary and secondary effects on air quality, human health and the environment. NO$_x$ is a toxic gas itself (WHO, 2003) and contributes to the formation of secondary pollutants and ozone.



Ozone close to the Earth's surface is a toxic pollutant which can lead to respiratory problems and has negative effects on plant growth and crop yield (e.g. Wang and Mauzerall (2004)). $NO_x$ also contributes to acid deposition and eutrophication, harming

sensitive ecosystems (European Environment Agency, 2019).

The international shipping sector is a strong source of $NO_x$ and other air pollutants to the atmosphere (eg. Eyring et al. (2010)). Previous studies suggest that international shipping makes up for annual emissions of 2.0-10.4 TgN (Crippa et al., 2018; Eyring et al., 2010; Johansson et al., 2017), or 15-35% of total anthropogenic $NO_x$ emissions worldwide. While cars, the power sector and industry have shown substantial reductions in their emissions over the last 10-20 years in Europe and the United States

(Curier et al., 2014; Hassler et al., 2016), $NO_x$ emissions from shipping activity have increased (De Ruyter De Wildt et al., 2012; Boersma et al., 2015) and the number of ship movements and ship size is expected to keep increasing in the future (Eyring et al., 2005; UNCTAD, 2019). Shipping-related air pollution emissions are estimated to lead to 60,000 premature deaths annually, especially in coastal regions (Corbett et al., 2007; Marais et al., 2015).

To mitigate these and other harmful impacts, more stringent regulations on $NO_x$ emissions for ships have been implemented

in coastal regions and on the open ocean (IMO, 2013). For example, ships built in 2011 or later have to follow Tier II nitrogen emission regulation as defined in MARPOL Annex VI. In so-called Emission Control Areas (ECAs) even more stringent rules apply. From 1 January 2021 onwards, the new MARPOL Annex VI Regulation 13 determines that newly built ship engines should be compliant with Tier III in the new ECA in the Baltic and North Sea, which should result in 75% lower $NO_x$ emissions from new ships. The exact limits depend on ship engine speed (IMO, 2013), see Supplement.

For new regulations to be effective, monitoring and verification of ship emissions is required. Traditional compliance monitoring includes national authorities conducting on-board checks of engine certificates and keel-laying date. This is not a direct verification of emissions and can only be done for a limited number of vessels. Other methods are on-board measurements at the ships exhaust pipe (e.g. Agrawal et al. (2008)) or downwind measurements of emission plumes using sniffer techniques or DOAS (Differential Optical Absorption Spectroscopy) measurements (e.g. Lack et al. (2009); Berg et al. (2012); McLaren et al.

(2012); Pirjola et al. (2014); Seyler et al. (2019)). Modern techniques also include airborne platforms such as helicopters, small aircrafts (Mellqvist and Conde, 2021; Chen et al., 2005b) or drones (van Roy and Scheldeman, 2016). While these methods do not require inspectors to enter the vessel, they require proximity to the ships monitored and are thus less fit-for-purpose when a large number of ships is to be checked, or on open sea away from land. For the above reasons monitoring by satellite remote sensing is expected to provide a useful alternative.

Satellite instruments have observed enhancements of $NO_2$ column densities over major shipping routes, e.g. from GOME (Beirle et al., 2004), SCIAMACHY (Richter et al., 2004) and OMI (Vinken et al., 2014b; Marmer et al., 2009). These satellite measurements have recently been continued with new observations from the TROPOMI (TROPOspheric Monitoring Instrument) sensor. With a pixel size of 3.5x5.5 $km^2$ TROPOMI provides a spatially more resolved evaluation of $NO_2$ pollution patterns compared to its predecessors GOME (40x320 $km^2$), SCIAMCHY (30x60 $km^2$) and OMI (13x25 $km^2$). Indeed, previous studies

demonstrated TROPOMI's capability to pinpoint emissions from the mining industry (Griffin et al., 2019), emissions patterns within cities (Beirle et al., 2019; Lorente et al., 2019), emissions along a gas pipeline in Siberia (van der A et al., 2020) and even from individual ships in the Mediterranean Sea (Georgoulias et al., 2020). Ding et al. (2020) used TROPOMI $NO_2$ columns





and inverse modelling to show $NO_x$ emission reductions during the COVID-19 lockdown over urban centers and regions with
strong maritime transport.

While the aforementioned studies demonstrate the large potential of TROPOMI and its high resolution, retrieval problems
remain. Validation studies (e.g. Griffin et al. (2019); Verhoelst et al. (2021)) suggest a 15%-40% low bias in TROPOMI tro-
posheric vertical $NO_2$ ($N_{v,trop}$) columns relative to independent in-situ and MAX-DOAS measurements. Cloud properties
present one of the leading sources of uncertainty in trace gas retrieval from space (Boersma et al., 2004; Lorente et al., 2017)
and cloud heights used until (and including) v1.3 of the operational TROPOMI retrieval algorithm have been suggested to be

biased low (Compernolle et al., 2021). To address this bias in cloud heights, the Royal Dutch Meteorological Institute (KNMI)
recently updated the FRESCO+ cloud retrieval by widening the spectral window, which is supposed to improve the sensitivity
to low clouds.

This study presents and assesses the impact of steps towards an improved monitoring of shipping $NO_2$ with TROPOMI. First,
we demonstrate TROPOMI's capability to detect ship emissions applying a typical data selection and compare it to OMI's. We

examine previous suggestions of improved retrieval sensitivity over sun glint scenes (Georgoulias et al., 2020). Additionally,
we evaluate the new FRESCO+wide cloud pressure retrieval in and its impact on the TROPOMI $NO_2$ columns in v1.4/2.1 of
the operational TROPOMI $NO_2$ algorithm. Based on our findings, we create a data set of historical TROPOMI $NO_2$ columns
consistent with the v1.4 data allowing for otherwise challenging trend analysis. We conclude with an application of our findings
to quantify the effects of the COVID-19 pandemic on ship pollution, an unique opportunity to assess the relationship between

the anticipated emission reductions and observed $NO_2$ columns.

## 2 Methods and Materials

### 2.1 TROPOMI and OMI $NO_2$ column measurements

The European TROPOMI (Veefkind et al., 2012) is on board the Sentinel-5-Precursor launched in October 2017. TROPOMI
has a push-broom design with a 2-D detector, which measures back-scattered radiation from the Earth's atmosphere for viewing

zenith angles up to 57°, in the spectral region from UV to short wave infrared. The instrument is equipped with a polarization
scrambler, simplifying the radiative transfer analysis. The width of the TROPOMI swath is about 2600 km, which results in
daily (near-)global coverage with about 25 million measurement points. In band 4, where $NO_2$ is retrieved, TROPOMI provides
450 measurements across-track, with a minimal width of 3.5 km.

The design of OMI is similar to that of TROPOMI, but OMI measures in a smaller spectral range (270-500 nm) (Levelt et al.,

85 2006, 2018). Another important difference is that OMI has only 60 across-track measurements, with the smallest pixels having
a width of 25 km. Along track, the resolution of TROPOMI is 7 km (5.5 km since August 2019), compared to 13 km for OMI.
Combined, the area of the smallest TROPOMI pixel is 19 $km^2$, while it is 325 $km^2$ for OMI, a factor of 17 improvement in
spatial resolution. Both instruments are in a sun-synchronous ascending orbit and have an equator overpass time of about 13:30
hrs local time.

90 To retrieve tropospheric $NO_2$ columns, TROPOMI uses a 3-step retrieval approach based on the DOAS (Differential Opti-



cal Absorption Spectroscopy, Platt and Stutz (2008)) technique: first the slant column density ($N_s$) is retrieved by spectral fitting of a modeled reflectance spectrum to the observed reflectances in the 405-465 nm window (van Geffen et al., 2021b; Van Geffen et al., 2020; Zara et al., 2018). In the second step, data assimilation in the global chemistry Transport Model 5 (TM5-MP) results in vertical $NO_2$ profiles that are then used to separate the stratospheric and tropospheric contribution to the slant columns (Van Geffen et al., 2020; Dirksen et al., 2011). In the last step, Air Mass Factors (AMFs) are calculated (Lorente et al., 2017) to translate the $N_s$ into vertical column densities ($N_v$). The AMF is calculated using the DAK radiative transfer model (de Haan et al., 1987; Stammes, 2001), and accounts for the viewing and solar geometry as well as surface properties and cloud effects. Cloud height information is retrieved with TROPOMI's FRESCO+ cloud algorithm (driven by the 761 and 765 nm $O_2$ absorption depth), and cloud fraction from the reflectance levels within the 405-465 nm $NO_2$ fitting window. Other input parameters to the TROPOMI AMF calculation are the surface albedo climatology (Kleipool et al. (2008), 0.5°x0.5°), a priori $NO_2$ profiles simulated with TM5-MP (Williams et al. (2017), 1°x1°) and terrain height from Global 3 km Digital Elevation Model (DEM_3KM).

The retrieval of tropospheric $NO_2$ columns ($N_{v,trop}$) from OMI (Boersma et al., 2018) proceeds along the same lines, and is therefore similar in many aspects. On the other hand, especially spatial resolution, signal-to-noise and the retrieval of cloud properties differ as highlighted in Table 1.

Clouds have several relevant effects on $NO_2$ retrieval. Clouds shield the lower part of the atmosphere which is most influenced by anthropogenic emissions including those from shipping. Therefore, data users are typically advised to consider scenes with cloud radiance fractions below 50% (Eskes et al., 2019). Initial validation of TROPOMI $NO_2$ v1.2/1.3 pointed out that the FRESCO+ algorithm retrieves cloud heights close to the surface heights, leading to overestimations in the TROPOMI $NO_2$ AMFs, and, consequently underestimations of the tropospheric $NO_2$ columns (Verhoelst et al., 2021). Accurate knowledge of cloud fraction and height is key for high quality trace gas column retrievals (e.g. Boersma et al. (2004)). A detailed description of the TROPOMI and OMI cloud algorithms and recent updates therein is given in the following subsection.

## 2.2 Improved TROPOMI FRESCO+, OMI and VIIRS cloud retrievals

FRESCO+ (Fast Retrieval Scheme for Clouds from the Oxygen A band) retrieves cloud pressures from the relative depth of $O_2$-A band measurements (Koelemeijer et al., 2001; Wang et al., 2008) using three spectral windows at 758-759 nm (continuum, no absorption), 760-761 nm (strong absorption) and 765-766 nm (moderate absorption). In the algorithm, clouds are assumed to be Lambertian reflectors with a fixed albedo of 0.8, consistent with assumptions for the $NO_2$ AMF calculation. The surface albedo assumed in the cloud pressure retrieval is from the GOME-2 minimum LER climatology at 758 & 772 nm (Tilstra et al., 2017), which is a potential source of uncertainty in the cloud pressure retrieval as the resolution and overpass time of GOME-2 is different from TROPOMI. FRESCO+ has been compared to other cloud data sets by Compernolle et al. (2021), who reported on tendencies in FRESCO+ to overestimate cloud pressures.

To address the high-bias in TROPOMI FRESCO+ cloud pressures, a new version of the FRESCO+ algorithm was introduced and implemented in the operational $NO_2$ retrieval with the introduction of TROPOMI v1.4 in December 2020. This version,





**Table 1.** Retrieval settings for the TROPOMI and OMI $NO_2$ retrievals used in this work.

| | | TROPOMI v1.2/1.3 | TROPOMI v1.4/2.1[1] | OMI QA4ECV |
|---|---|---|---|---|
| | Data availability | ESA science hub | ESA science hub | qa4ecv.eu |
| | Public data period | 30 Apr 2018 – 29 Nov 2020 | 29 Nov 2020 - 01 Jul 2021 | Oct 2004 - |
| | Spatial resolution at nadir | 3.5 km × 5.5 km | 3.5 km × 5.5 km (3.5 km × 7 km) | 13 km × 25 km |
| $N_s$ | Fitting window | 405-465 nm | 405-465 nm | 405-465 nm |
| | Signal-to-noise ratio | ~1500 | ~1500 | ~500 |
| | Solar reference spectrum | Daily | Daily | 2005 average |
| | DOAS polynomial degree | 5 | 5 | 4 |
| | Intensity offset correction | no | no | yes |
| | Destriping | yes (since v1.2) | yes | yes |
| AMF | Surface albedo | OMI minimum LER at 440 nm (0.5°) | OMI minimum LER at 440 nm (0.5°) | OMI minimum LER at 440 nm (0.5°) |
| | A priori $NO_2$ profiles | TM5-MP at 1° × 1° | TM5-MP at 1° × 1° | TM5-MP at 1° × 1° |
| | Cloud retrieval | FRESCO+ | FRESCO+wide | OMCLDO2 |
| | Cloud fraction | Retrieved from 405-465 nm continuum | Retrieved from 405-465 nm continuum | Retrieved from 470-490 nm continuum |
| | Cloud pressure | Narrow $O_2$-A band (758, 761 and 765 nm) | Wide $O_2$-A band (758, 761 nm and 765-770 nm) | $O_2$-$O_2$ aborption feature (477 nm) |
| | Surface albedo in cloud pressure retrieval | GOME-2 minimum LER at 758 & 772 nm (0.25° × 0.25°) | GOME-2 minimum LER at 758 & 772 nm (0.25° × 0.25°) | OMI minimum LER at 758 & 772 nm (0.5° × 0.5°) |

125 called FRESCO+wide, uses a wider spectral window for the cloud retrieval (765-770 nm, see Table 1), which includes the flank of the absorption band, where oxygen absorption is weaker than in the center of the $O_2$-A band (761 and 765 nm). Adding weaker $O_2$ absorption features improves the sensitivity to clouds low in the atmosphere. This is not possible from the strong $O_2$ absorption at 761 nm, which is so close to saturation that it becomes difficult to use its absorption depth in order to distinguish between bright reflecting layers at the Earth's surface from reflecting surfaces in the lower atmosphere.

130 Prior to the implementation of FRESCO+wide in the operational TROPOMI $NO_2$ retrieval in December 2020, KNMI produced

---

[1]In addition to improved cloud parameters, TROPOMI v2.1 data has improved further through a better calibration of level-1 spectra, especially in the treatment of outliers and saturation (Ludewig et al., 2020), and through improvements in the NO2 algorithm itself (van Geffen et al., 2021a). Version v2.1 is only used for production of the DDS-2B test data, not for publicly released data. Version v2.2, available publicly as of July 2021, is essentially the same as v2.1.



4 periods with TROPOMI $NO_2$ test data based on FRESCO+wide, the so-called diagnostic data set 2B (DDS-2B). DDS-2B contains data from four v1.2/v1.3 periods during 2018-2019 additionally processed with v2.1 of the TROPOMI algorithm. The most significant difference between the two is that v2.1 (and v1.4) uses cloud fractions and AMFs determined from the FRESCO+wide cloud pressure instead of the FRESCO+ cloud pressures used in v1.2/v1.3 data.

Additionally, we use co-sampled cloud information from the Ozone Monitoring Instrument (OMI) on board of EOS-Aura. The OMI OMCLDO2-retrieval uses the relative depth of the $O_2$-$O_2$ absorption feature at 477 nm to retrieve cloud pressures (Acarreta et al., 2004; Veefkind et al., 2016). The general approach of using Lambertian reflectors is similar to the FRESCO+ algorithm, but an important difference is that the OMCLDO2-algorithm needs to account for Raman scattering and $O_3$ absorption, and that the absorption strength of the $O_2$-$O_2$ features is proportional to the square of the $O_2$ concentration, making it

more sensitive to low clouds compared to FRESCO+.

We also use cloud information from VIIRS (Visible/Infrared Imager/Radiometer Suite) on board of the SUOMI National Polar-orbiting Partnership (SNPP) as a completely independent means of verification. SNPP orbits the Earth in a sun-synchronous, ascending node with full daily global coverage and observes the same scenes as TROPOMI within three minutes. We use NASA's CLDPROP L2 VIIRS-SNPP cloud product (Platnick et al., 2017) with a resolution of 750 m at nadir, which provides

a cloud mask, cloud (top) pressure, cloud optical thickness (COT) and cloud water phase for each pixel retrieved. The VIIRS retrieval derives a cloud top temperature using an optimal estimation approach in the thermal infrared spectral bands M14-M16 (8.5-12.3$\mu$m). In a subsequent step, these cloud top temperatures are converted to cloud pressures using Numerical Weather Prediction temperature profiles (Heidinger and Li, 2017). In addition to the cloud top pressure, we use the VIIRS cloud optical thickness (COT) to generate (effective) cloud fractions that can be compared directly to the TROPOMI cloud fractions. First,

we derive a geometrical cloud fraction by calculating the share of cloudy VIIRS pixels per grid cell. Then, we translate this geometrical cloud fraction $f_{c,geo}$ into a effective cloud fraction $f_{c,eff}$ using:

$$f_{c,eff} = f_{c,geo} * a_c/0.8 \tag{1}$$

with $a_c$ the cloud albedo. The cloud albedo is calculated from the VIIRS COT and a previously established empirical relationship between cloud optical thickness and cloud albedo for liquid water clouds (Buriez, 2005; Boersma et al., 2016).

To evaluate the improvements in the FRESCO+wide retrieval, we compare daily gridded, co-sampled cloud data from (partly) cloudy pixels seen by TROPOMI (FRESCO+ and FRESCO+wide), OMI and VIIRS over parts of the Mediterranean Sea (37.0°N-41.25°N, 2.0°W-8.0°W), the Bay of Biscay (43.5°N-47.5°N,10.0°E-3.0°E) and Northwestern Europe (50.0°N-53.0°N, 4.0°W-9.0°W). These regions represent different surface types (land and ocean), climatological conditions and pollution levels. We define partly cloudy pixels as all pixels with an effective cloud fraction $f_c \geq 0.05$. For TROPOMI we additionally apply

sufficient quality of retrieval ($qa \geq 0.5$) and a pressure difference between surface pressure and cloud pressure of at least 7 hPa. The last filter is applied to filter out 'ghost' clouds coming from sun glint viewing geometries (see Sec. 2.3 below). For OMI, we use the OMCLDO2 cloud properties and take only pixels with solar and viewing zenith angle smaller than 80° into account.



As Eq. 1 is valid for liquid water clouds only, we select liquid water clouds, and reject ice clouds, as indicated by the VIIRS cloud water phase.

### 2.3 Sun glint in the TROPOMI NO$_2$ retrieval

The term *sun glint* refers to particular satellite viewing geometries, under which the ocean acts as a mirror by reflecting sun light directly to the satellite instrument. In the TROPOMI data product pixels that are potentially in sun glint mode are identified based on the combination of their solar and viewing zenith and azimuth angles. The sun glint condition is fulfilled when the scattering angle $\Theta$ is smaller than a threshold angle $\Theta_{\mathrm{max}}$:

$$\Theta = \arccos\left[\cos\theta\cos\theta_0 - \sin\theta\sin\theta_0\cos\left(\phi_0 - \phi\right)\right] \leq \Theta_{\mathrm{max}} \tag{2}$$

with $\theta$ and $\theta_0$ the solar and viewing zenith angles and $\phi$ and $\phi_0$ the solar and viewing azimuth angles, respectively. For the TROPOMI data products the maximum threshold angle has been set at 30°. Smaller angles are used before, e.g. for SCIAMACHY and GOME-2 (Loots et al., 2017). The TROPOMI algorithm treats the enhanced albedo as a partially cloudy scene with the cloud pressure located at or close to the sea surface.

### 2.4 Relationship between NO$_x$ emissions and columns

When studying NO$_2$ columns to investigate emission trends, the non-linearity of NO$_x$ chemistry needs to be taken into account. For example, the lifetime of NO$_x$ depends on the background O$_3$ level, the available sun light and NO$_x$ concentrations themselves (Jacob, 1999). We use a (modeled) $\beta$ factor to express the sensitivity of relative NO$_2$ column changes to changes in the relative emission strength following the approach in Vinken et al. (2014a) and Verstraeten et al. (2015) with

$$\beta = \frac{\Delta E/E}{\Delta N/N} \tag{3}$$

where $\Delta E/E$ represents the imposed relative change in NO$_x$ emission flux and $\Delta N/N$ the relative change in subsequently simulated tropospheric NO$_2$ columns. Here, we use monthly $\beta$ values from Verstraeten et al. (2015), where the sensitivity of tropospheric NO$_2$ to a 15% increase in emissions has been modelled with TM5 on a 3°x2°grid for 2006. As we are interested in European Seas only, we average $\beta$ in the area 35°N-40°N and 5°E-10°W for Gibraltar and 30°N-37°N and 15°W-35°W for the Eastern Mediterranean. We use the resulting $\beta$ value to estimate relative changes in NO$_x$ emissions $(E_{2020} - E_{2019})/E_{2019}$ as

$$\frac{E_{2020} - E_{2019}}{E_{2019}} = \beta \cdot \frac{N_{obs,2020} - N_{obs,2019}}{N_{obs,2019}} \tag{4}$$

where $(N_{obs,2020} - N_{obs,2019})/N_{obs,2019}$ is the observed relative change in NO$_2$ columns.

### 2.5 AIS data and ship specific data

To relate the TROPOMI NO$_2$ columns to shipping activity, we use data from the Automatic Identification System (AIS) for shipping. Since 2005, the International Maritime Organization (IMO) requires all ships with a gross tonnage over 300 and all



passenger ships to carry an AIS transponder. These transponders broadcast static (e.g. identity, size) and dynamic (e.g. position, speed, course) information of the ship, which can be received by other ships, shore stations, and satellites (International Maritime Organization (IMO), 2014). Here we use historical AIS data available to the Dutch Human Environment and Transport

Inspectorate (ILT) to assess changes in shipping activity over densely travelled European shipping lanes in 2019 and 2020. We use AIS data of ships in a part of the shipping lane in the Eastern Mediterranean (31.91°N-34.53°N and 25.91°E-27.67°E) and close to the Strait of Gibraltar ( 35.0°N-37.0°N and 4.0°W-2.5°W). Furthermore, we use information on ship dimensions from the official ship registrations[2] to calculate a ship emission proxy $E$ from ship length $L$ and ship speed $v$ as $E = L^2 \cdot v^3$ as used e.g. in Georgoulias et al. (2020). For the areas and times under study, ship specific data was available only for 50% (Gibraltar)

and 70% (Eastern Mediterranean) of the ships.

## 3   Results

We start with demonstrating TROPOMI's capabilities to detect shipping $NO_2$ applying established data selection criteria. Next, we show steps to optimize monitoring of ship emissions making use of sun glint (Sec. 3.2) and recent improvements in the cloud retrieval (Sec. 3.3) and compare the improved TROPOMI data to OMI data in Sec. 3.4. We end with an application of

our findings to quantify $NO_2$ emissions reductions from shipping due to COVID in 2020 in Sec. 3.5.

### 3.1   Detection of $NO_2$ pollution over European shipping lanes

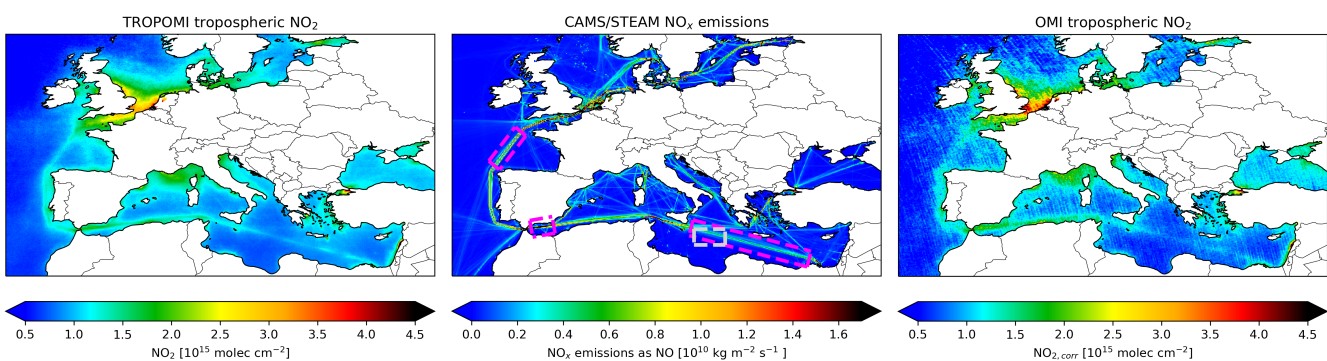

**Figure 1.** Summertime (May-September) mean tropospheric $NO_2$ columns from TROPOMI (left panel) and OMI (right panel) over European seas in 2019. The center panel shows the summertime mean $NO_x$ emissions from the CAMS/STEAM emission inventory (Granier et al., 2019; Johansson et al., 2017). The gray and pink rectangles in the center panel indicate areas used in Sec. 3.2 and Sec. 3.4, respectively.

TROPOMI detects unprecedented spatial detail in shipping $NO_2$ over busy shipping routes. Fig. 1 shows the summertime mean (May-September 2019) $NO_2$ columns from TROPOMI and OMI averaged to a common 0.0625°x0.0625° grid. We find a clear signal of shipping $NO_2$ west of Portugal and from the Strait of Gibraltar to the East. There are further indications of enhanced

---

[2] gisis.imo.org

**Figure 2.** 2019 summertime mean (May-September) tropospheric NO$_2$ columns from TROPOMI (left panel) and summertime mean NO$_x$ emissions from the CAMS/STEAM emission inventory (right, Granier et al. (2019); Johansson et al. (2017)) of shipping lanes around Denmark (top) and Eastern Aegeean Sea (bottom) for the first time detected with satellites.

NO$_2$ related to shipping in the Bay of Biscay from the tip of Brittany towards the North-West of Spain, and in the Eastern Mediterranean from South of Sicily towards the Suez Canal. Previous studies reported NO$_2$ enhancements over these shipping lanes with other satellites (e.g. by OMI (Vinken et al., 2014b)). Additionally, we see a clear NO$_2$ enhancement in the Aegean Sea between Istanbul and the Greek Islands as well as around Denmark as shown in Fig. 2, which to the authors' knowledge have not been observed by satellite instruments previously. Furthermore, (clear) hints of shipping activity can be seen in the





Baltic Sea, the Eastern Aegean Sea, the Adrian Sea, north-east of Corsica, the British Channel, and several forks in the Eastern Mediterranean and south east of Sicily, which are all present as shipping lanes in CAMS/STEAM emissions (Granier et al., 2019; Johansson et al., 2017). Corresponding zoomed in maps of TROPOMI tropospheric $NO_2$ and CAMS/STEAM emissions are shown in the Appendix A. For the analysis, we selected mostly clear-sky pixels with a quality assurance value (qa) of 0.75 or higher as recommended in the TROPOMI (Eskes et al., 2019) and (equivalent settings) OMI user manuals (Boersma et al.,

2017). These enhancements are not an artefact in the retrieval coming from the AMF calculation (see Section 2.1) as they are visible in geometric tropospheric vertical column densities $N_{trop,geo}$ using a geometric AMF (see Appendix B) shown in Fig. B1.

TROPOMI and OMI show a comparable high spatial correlation to CAMS/STEAM emission data of $R = 0.93$ and $R = 0.91$, respectively. For the calculation, we brought OMI and TROPOMI troposhperic data to the CAMS resolution (0.1°x0.1°)

and selected only grid cells over the Mediterranean Sea. This was done to ensure comparable meteorological and chemical conditions. Next, we binned the data by emission strength in bins of $0.05 \cdot 10^{-10}$ kg m$^{-2}$ s$^{-1}$. A reduced major axis regression of all bins with more than 10 entries lead to the correlation coefficients given above. Corresponding scatter plots can be found in Fig. C1. The y-axis intercept of $1.07$ (1.05) $\cdot 10^{-15}$ molec·cm$^{-2}$ for TROPOMI (OMI) represents the mean background $NO_2$ column over the summertime Mediterranean. Other emission bin sizes lead to slightly different but comparable regression

results.

Besides the higher resolution of the TROPOMI instrument, TROPOMI $N_{v,trop}$ thus have a comparable spatial correlation with emission inventories when compared to OMI's. The distinct shipping lanes visible in Fig. 1 and B1 visualize TROPOMI's unprecedented capabilities to detect shipping $NO_2$.

## 3.2 Sun glint

For situations of sun glint (see Sec. 2.3) the usually dark ocean appears bright in the TROPOMI data, leading to a strong increase in the effective scene albedo with decreasing scattering angle as shown in Fig. 3(a). Figure 3(b) shows that the increase in scene albedo leads to substantially higher vertical sensitivities, as diagnosed by the averaging kernels (AK) in the operational TROPOMI $NO_2$ product. The sensitivity increased most in the lowest vertical layer, where the kernel values are on average ≈60% higher for sun glint compared to non sun glint circumstances (0.44 vs 0.28). Increased albedo generally

enhances a satellite sensor's sensitivity to $NO_2$ concentrations in the lower atmosphere (e.g. Eskes and Boersma (2003)), and sun glint scenes have been tentatively used previously to attribute shipping plumes to individual ships in the Mediterranean Sea (Georgoulias et al., 2020).

The scene albedo and vertical sensitivity can be further increased by focusing on scenes with low-moderate wind speeds (≈ 2 m/s). Fig. 4(a) shows the relationship between effective scene albedo and wind speed for scenes with small scattering angles

$\Theta \leq 15°$. For very low wind speeds the mean scene albedo is almost as small as for non sun glint scenes and smaller than for all other wind speeds. For wind speeds between 1.5 and 2.0 m/s we find an effective scene albedo of almost 0.25, which is approximately double compared to the average for these scattering angles and more than 5 times as high as for non sun glint scenes. For higher wind speeds the scene albedo decreases to around 0.10. In Fig. 4(b) the effect on the averaging kernel profile





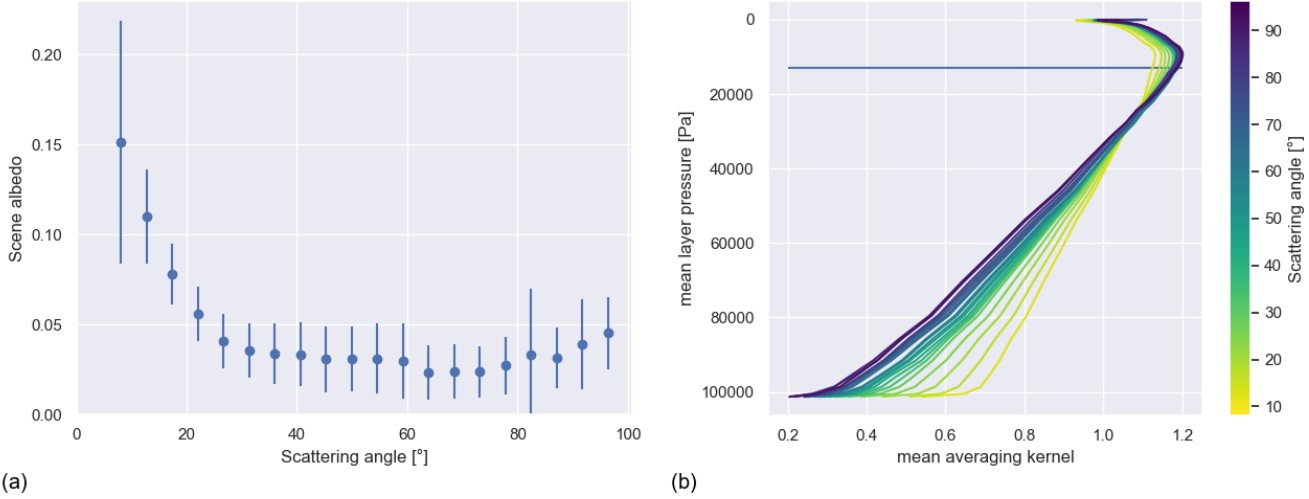

**Figure 3.** (a) Change of effective scene albedo with scattering angle over the Central Mediterranean north of Libya in June-July-August 2018 (see gray rectangle in Fig. 1, $\approx$200,000 data points in total). The error bars indicate the standard deviation of each bin. (b) Mean averaging kernel (profiles) for different scattering angles, sampled as for (a). Only pixels with cloud radiance fractions $< 0.25$ or $p_{surf} - p_{cloud} \leq 300\,\mathrm{Pa}$ were selected. The blue line in (b) indicates the average tropopause altitude.

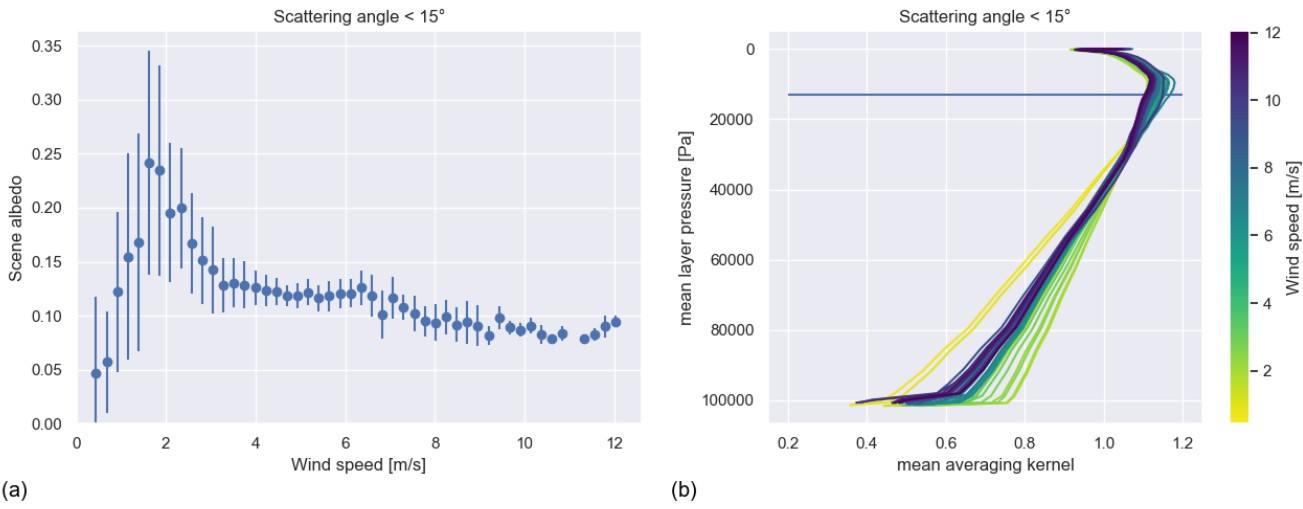

**Figure 4.** (a) Change of effective scene albedo with wind speed over the Central Mediterranean north of Libya in June-July-August 2018 (see gray rectangle in Fig. 1) for scenes with scattering angles smaller than 15° ($\approx$22,000 data points in total). The error bars indicate the standard deviation of each bin. (b) Mean averaging kernel (profiles) for different wind speeds, sampled as for (a). Only pixels with cloud radiance fractions $< 0.25$ or $p_{surf} - p_{cloud} \leq 300\,\mathrm{Pa}$ were selected. The horizontal blue line in (b) indicates the average tropopause altitude.





is shown. As expected low wind speeds lead to the smallest AK in the lower atmosphere, whereas wind speeds between 1.5 and
2.0 m/s show the largest AKs close to the sea surface. This relationship can be understood in terms of wind-induced sea surface
roughness (Cox and Munk, 1956). Both very low and strong winds limit the probability that a scattering angle $\Theta \leq \Theta_{max}$ leads
to sun glint effects at the sensor: For very low wind speeds, the sea surface is effectively flat, leading to sun glint only for very
small scattering angles $\Theta \ll \Theta_{max}$, whereas for strong winds the sea surface is so rough that the sun light is reflected in all
directions, making the reflections towards the satellite instrument unlikely.

Additionally, we find that sun glint scenes can be used with confidence for detecting ship pollution signals from UV/Vis
spectrometers such as TROPOMI and the usage of sun glint data should be encouraged. The (normalized) tropospheric slant
columns ($N_{trop,geo} = N_{s,trop}/M_{geo}$) observed under sun glint conditions are 20-25% higher than under non sun glint condi-
tions as shown in Fig. 5(a). Vertical profiles of $NO_2$ over oceans typically feature enhancements from ships within the marine
boundary layer, and small background levels above (e.g. Chen et al. (2005a); Boersma et al. (2008), see Fig. 1). Therefore, it is
no surprise that the AK increases in the lower atmosphere lead to small but detectable increases in (tropospheric) slant columns
over the study region covering a frequently travelled shipping lane.

The enhanced slant columns are correctly accounted for by increased AKs leading to reliable retrievals under sun glint. Fig. 5(b)
compares the tropospheric vertical columns reported in the official TROPOMI $NO_2$ product sampled under sun glint compared
to non sun glint conditions. The differences between the distributions are only small. Mean values for scene albedo, (nor-
malized) tropospheric slant columns and tropospheric vertical columns reported in the official TROPOMI $NO_2$ product for
different scattering angles are summarized in Table 2.

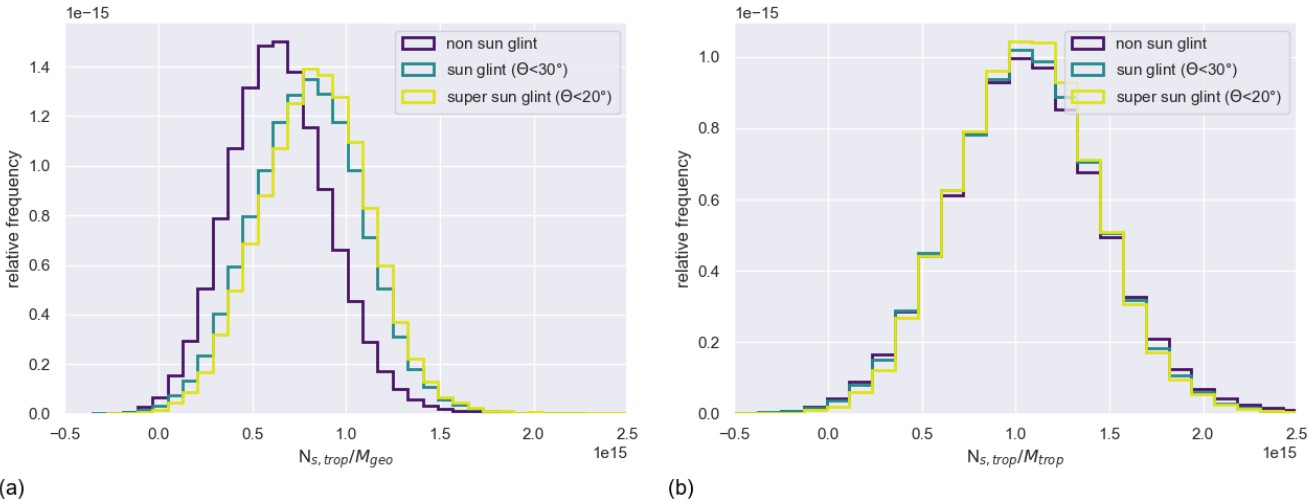

**Figure 5.** (a) Probability distribution of tropospheric $NO_2$ columns ($N_{s,trop}/M_{geo}$) over the Central Mediterranean north of Libya in June-
July-August 2018 (see grey rectangle in Fig. 1) taken under non sun glint, sun glint ($\Theta \leq 30°$) and super sun glint ($\Theta \leq 20°$). (b) Probability
distribution of tropospheric $NO_2$ columns in the official TROPOMI $NO_2$ product ($N_{s,trop}/M_{trop}$) for the same data selection.



**Table 2.** Summary of mean effective scene albedo, normalized tropospheric slant column and $N_{v,trop}$ for non sun glint, sun glint ($\Theta \leq 30°$) and super sun glint ($\Theta \leq 20°$) over the Central Mediterranean north of Libya in June-July-August 2018 (see gray rectangle in Fig. 1).

|  | non sun glint | sun glint | super sun glint |
|---|---|---|---|
| Effective scene albedo | $0.03 \pm 0.02$ | $0.08 \pm 0.05$ | $0.11 \pm 0.05$ |
| $N_{trop,geo}$ (molec·cm$^{-2}$) | $(0.65 \pm 0.28) \cdot 10^{15}$ | $(0.80 \pm 0.30) \cdot 10^{15}$ | $(0.83 \pm 0.30) \cdot 10^{15}$ |
| $N_{v,trop}$ (molec·cm$^{-2}$) | $(1.06 \pm 0.42) \cdot 10^{15}$ | $(1.05 \pm 0.40) \cdot 10^{15}$ | $(1.06 \pm 0.38) \cdot 10^{15}$ |

### 3.3 Cloud properties

Here we evaluate TROPOMI's capability to retrieve realistic cloud parameters retrieved from the 405-465 nm continuum reflectances and effective cloud pressures from the $O_2$-A band (Table 1), addressing recent improvements in the FRESCO+ algorithm to avoid overestimated cloud pressures (Compernolle et al., 2021). These improvements in cloud retrievals lead to an inconsistency in the tropospheric $NO_2$ column record.

### 3.3.1 Cloud fractions

We find that improved TROPOMI cloud fractions are of sufficient quality to support the TROPOMI $NO_2$ AMF calculation. They show good correlation to independent data such as from OMI and VIIRS. TROPOMI v1.2 and v2.1 cloud fractions are very similar with the new v2.1 cloud fractions being slightly smaller. More details can be found in Appendix D1.

### 3.3.2 Cloud pressure

FRESCO+wide cloud pressures are a clear improvement over the FRESCO+ data used in v1.2/1.3. Figure 6 shows a comparison of gridded, co-sampled cloud pressure distributions from TROPOMI v1.2 (FRESCO+), TROPOMI v2.1 (FRESCO+wide), OMI QA4ECV and VIIRS over the Bay of Biscay between 1 and 7 July 2018. As expected, the improved TROPOMI v2.1 cloud pressures are $\approx 40$ hPa lower than for v1.2, in line with their enhanced sensitivity, and show more realistic, elevated clouds. It is apparent that OMI cloud pressures are generally lower and show a flatter distribution than the other products. TROPOMI v2.1 and v1.2 show similar distributions as VIIRS, with v1.2 pressures higher by 50 hPa in the median, and v2.1 moving closer to VIIRS with a difference of 2 hPa relative to VIIRS. We find similar agreement between TROPOMI and independent data over the Mediterrenean Sea and northwestern Europe as shown in Table D2. FRESCO+wide cloud pressures agree best but remain higher than VIIRS in the median (both FRESCO cloud pressure distributions show a larger tail towards low pressures compared to VIIRS, possibly caused by filtering for liquid water clouds in VIIRS) . This is in line with expectations as VIIRS's infrared cloud retrieval is mostly sensitive to the cloud top (Platnick et al., 2017), whereas FRESCO's $O_2$-A band retrieval is more sensitive to the center of a cloud (e.g. Sneep et al. (2008)).



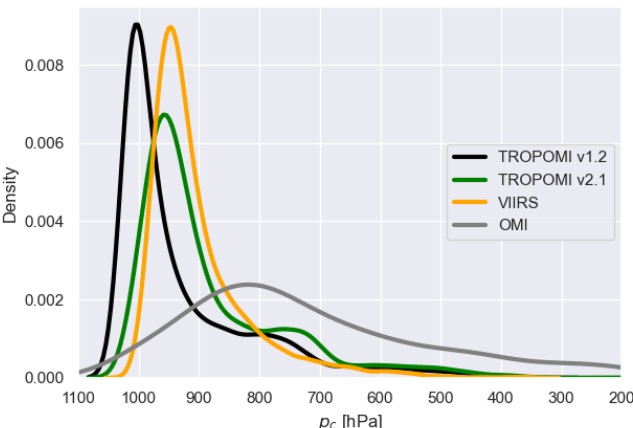

**Figure 6.** Probability distribution function of effective cloud pressures from TROPOMI v1.2, TROPOMI v2.1, OMI, and VIIRS for 1-6 July 2018 over the Bay of Biscay. Only cloud pressures for cloud fractions between 0.05 and 0.20 were selected, as these are most relevant for AMF calculations for mostly clear-sky pixels.

**Table 3.** Evaluation of TROPOMI v2.1 cloud pressures against reference data for the Bay of Biscay.

|  | Median cloud pressure [hPa] | 10th/90th percentile [hPa] | Geometric mean [hPa] |
|---|---|---|---|
| TROPOMI v1.2 | 979 | 753/1017 | 925 |
| TROPOMI v2.1 | 930 | 717/988 | 884 |
| OMI QA4ECV | 769 | 426/934 | 720 |
| VIIRS | 928 | 794/973 | 901 |

### 3.3.3  Effect of improved cloud pressure on TROPOMI NO$_2$ columns

The improved cloud pressures lead to increases of NO$_2$ columns of up to 40% depending on area and season. The left panel of Fig. 7 shows the change in tropospheric NO$_2$ columns as a function of cloud pressure over the Bay of Biscay and northwestern Europe in Summer. We see that NO$_2$ columns increase most for locations that had the highest original v1.2 cloud pressures, and that the improvements are strongest when cloud pressures are reduced most (light blue dots). The increase over the Bay of Biscay is smaller (up to $0.1 \cdot 10^{15}\ molec \cdot cm^{-2}$) than over northwestern Europe (up to $1.0 \cdot 10^{15}\ molec \cdot cm^{-2}$), reflecting

the higher pollution levels over the mainland. We see similar patterns with stronger improvements in Winter, as shown in the right panel of Fig. 7. The increased v2.1 NO$_2$ columns indicate that the v1.2 TROPOMI NO$_2$ product suffers from a 'cloud shielding' effect: NO$_2$ columns are underestimated due to too low clouds situated within the polluted boundary layer and that improved v2.1 cloud pressures (at least partly) resolve the low bias in v1.2 NO$_2$ columns. For this analysis, we compared the TROPOMI v2.1 columns retrieved with improved cloud information, to the TROPOMI v1.2 NO$_2$ columns. We used 10 days



in 4 different seasons[3] for which both v2.1 test data and v1.2 operational data were available to us as part of the DDS-2B. Our comparison focused on mostly clear-sky situations ($f_c < 0.2$), which are most relevant for detection of near-surface pollution sources.

**Figure 7.** Difference between tropospheric $NO_2$ columns retrieved with TROPOMI v1.2 and v2.1 as a function of the v1.2 cloud pressure for 27 June – 6 July 2018 over the mildly polluted Bay of Biscay in summer (a) and winter (b) as well as and the highly polluted Northwest Europe (c & d). Colors indicate the difference in cloud pressures between the two versions. The marker size is proportional to the logarithm of the sample size and the black line shows the effective average.

We trained a Deep Neural Network (DNN) to to predict v2.1 columns for the full TROPOMI mission period up to December 2020 and thereby created a consistent data set. The DNN-predicted v2.1 (hereafter v2.1p) reduces the mean difference to the

---

[3]27 June – 6 July 2018, 28 December 2018 – 5 January 2019, 25 March – 5 April 2019, and 13 - 23 September 2019





retrieved v2.1 NO$_2$ columns to $< 0.01 \cdot 10^{15}\ molec \cdot cm^{-2}$ (original v2.1 – v1.2 mean difference was $0.12 \cdot 10^{15}\ molec \cdot cm^{-2}$) over the 3 areas (see Sec. 2.2) of study during the 4 periods, suggesting considerable skill in the DNN approach. Details can be found in Appendix E.

Figure 8 shows the averaged NO$_2$ columns from v2.1p over the Summer of 2019 and Winter of 2019/20. The difference map in the right panel indicates that predicted v2.1 NO$_2$ columns are higher by up to $0.5 \cdot 10^{15}\ molec \cdot cm^{-2}$, especially over the

most polluted seas such as the English Channel and shipping lanes. We find a stronger impact of the improved cloud pressures in the winter season, reflecting that NO$_2$ pollution is confined in a thinner marine boundary layer in that season.

### 3.4 Comparison of TROPOMI and OMI NO$_2$ columns in shipping lanes

TROPOMI detects a more pronounced and narrower region of ship NO$_2$ pollution than OMI. On average, TROPOMI v2.1p

detects 45% higher peak NO$_2$ values than OMI. TROPOMI data allow the attribution of 14% more NO$_2$ to shipping lane enhancements, over 23% narrower shipping lanes. To quantitatively compare TROPOMI's capability to detect NO$_2$ over shipping lanes under different measurements conditions and compare it to OMI's, we created average NO$_2$ cross sections over busy shipping lanes. We studied NO$_2$ enhancements in summer 2019 (June-August) over shipping lanes in the Bay of Biscay, from Sicily to the Suez Canal, and East of Gibraltar, the regions visually defined in Fig. 1. First, we defined the location of the ship-

ping lanes according to the emission data shown in Fig. 1. Then, we calculated the average NO$_2$ columns along the shipping lane and parallel to it, taking care to exclude NO$_2$ columns measured over land. In that way we created an average cross section of NO$_2$ over shipping lanes. In the last step, we performed a background correction by subtracting a linear NO$_2$ background to isolate the NO$_2$ enhancements caused by shipping. The orbital data was gridded to regular grids of 0.0625°x0.0625° and 0.125°x0.125° resolution for TROPOMI and OMI, respectively. For TROPOMI only pixels with qa $> 0.75$ were taken into

account. For OMI, a consistent filtering was applied, including maximal solar and viewing zenith angles of 80° and maximal cloud radiance fractions of 0.5. The resulting cross sections are shown in Fig. 9. Table 4 summarizes the peak value, the area under the curve (i.e. the total NO$_2$ attributed to shipping) and the full width at half maximum (FWHM) for the three shipping lanes. It should be noted that the grid used for OMI is 2x coarser than the one used for TROPOMI. Gridding TROPOMI to the coarser grid used for OMI only changes the results slightly, indicating that the improved spatial resolution of TROPOMI

indeed improves the detection of NO$_2$ from narrow ship lanes and is in line with the finding of new shipping lanes shown Fig. 2.

As already seen in Fig. 8, the v2.1p data set shows slightly higher NO$_2$ compared to the TROPOMI v1.2/v1.3 data, especially in the center of the lane while background NO$_2$ is less affected by the correction. The impact of the DNN is larger in winter than in summer as discussed before.

For the Bay of Biscay it is also apparent that the NO$_2$ peak is shifted to the East for all data sets. As the location is defined by an emission inventory based on AIS data (and therefore real ship location), this is likely an effect of dominant westerly winds. We conclude that TROPOMI provides a significant improvement for the detection of shipping NO$_2$ with sharper and more pronounced shipping lanes in seasonal averages. The improved v2.1p TROPOMI data increase the signal further.

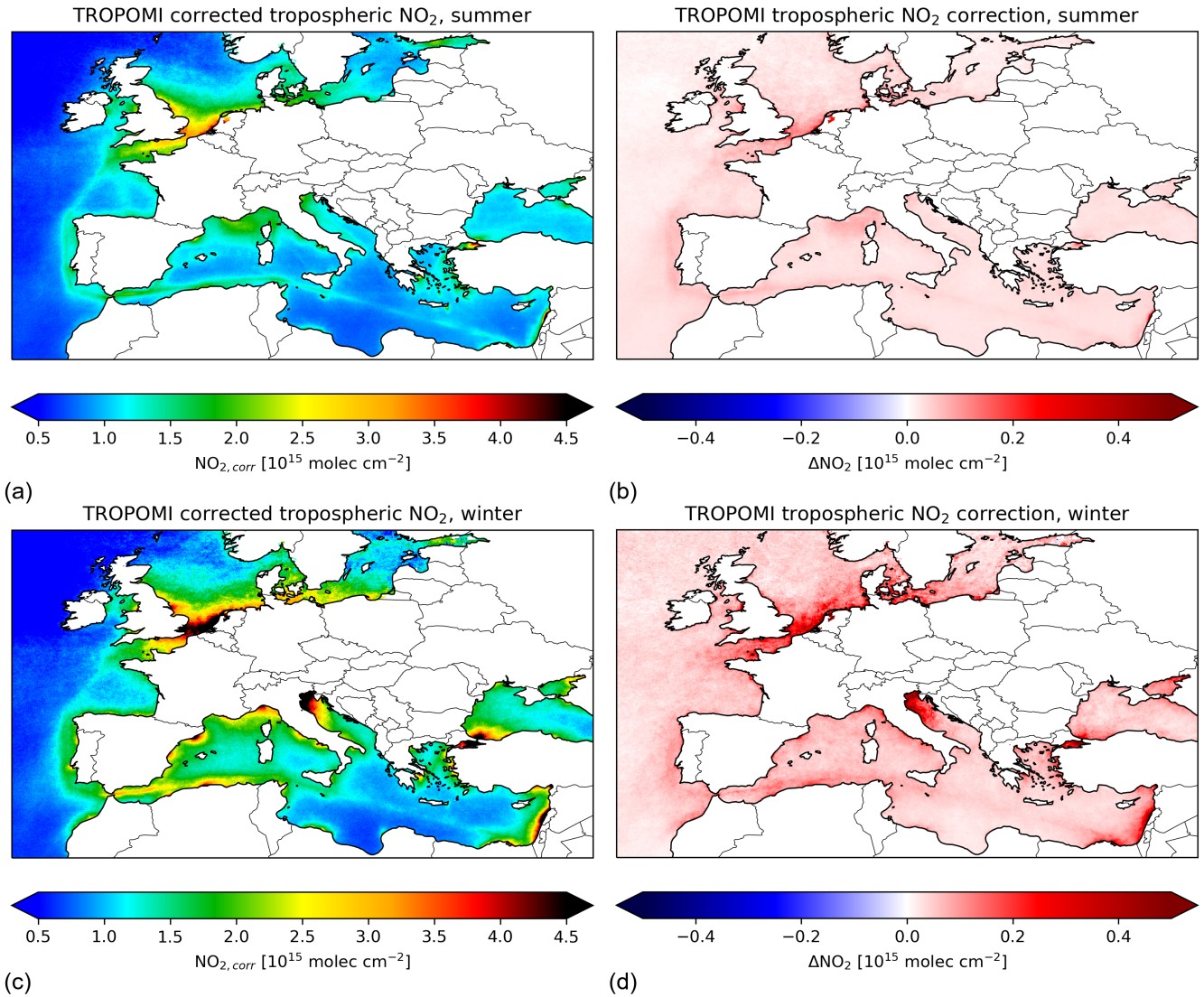

**Figure 8.** Effect of DNN correction: (a) Corrected TROPOMI data for summer (May-September) 2019, (b) change in $NO_2$ columns by the correction for the same period, (c) Corrected TROPOMI data for winter (November-April) 2019/2020, (d) change in $NO_2$ columns by the correction for the same period. Land areas are whitened out for clearness.

## 3.5 Covid

Emissions proxies derived from AIS data and from TROPOMI $NO_2$ suggest emission reductions from shipping in 2020 compared to 2019. While in the first three months of 2020 the emissions were generally higher compared to 2019, both emission proxies show reductions starting in April and lasting until the end of the year. This reduction can be linked to the COVID-19 pandemic, which led to economic lockdowns in many countries of the world. Europe had its most stringent measures in Spring





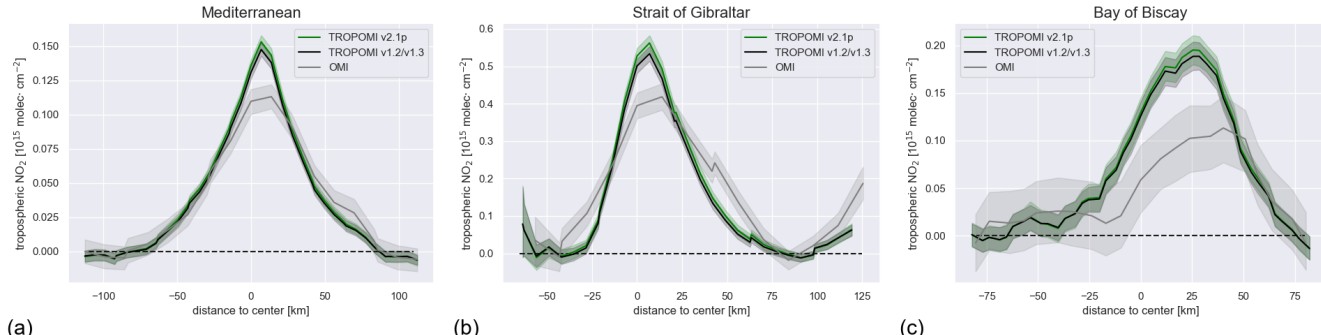

**Figure 9.** Mean enhancement cross sections in June-August 2019. TROPOMI v1.2/v1.3 in black, the improved TROPOMI v2.1p in green and OMI in grey. Shaded areas indicate the 95% confidence interval.

**Table 4.** Statistics for the $NO_2$ enhancement cross section over the Mediterranean shipping lane.

| Shipping lane | Product | max [$10^{15}$molec·cm$^{-2}$] | area under curve [$10^{15}$km·molec·cm$^{-2}$] | FWHM [km] |
|---|---|---|---|---|
| Bay of Biscay | TROPOMI v1.2/v1.3 | 0.189 | 11.34 | 54.6 |
| | TROPOMI v2.1p | 0.195 | 11.72 | 54.6 |
| | OMI | 0.113 | 7.75 | 59.2 |
| Strait of Gibraltar | TROPOMI v1.2/v1.3 | 0.534 | 24.47 | 41.3 |
| | TROPOMI v2.1p | 0.562 | 26.18 | 41.7 |
| | OMI | 0.418 | 28.83 | 63.8 |
| Mediterranean | TROPOMI v1.2/v1.3 | 0.148 | 8.80 | 52.3 |
| | TROPOMI v2.1p | 0.153 | 9.19 | 52.6 |
| | OMI | 0.113 | 8.56 | 70.2 |

and Autumn 2020.

We created daily 0.0625·0.0625° maps of TROPOMI data, using v2.1p $NO_2$ columns as described in Section 3.3.3 with qa $\geq$ 0.75. We calculate the area under the cross section as a measure for shipping $NO_2$ for monthly mean $NO_2$ columns for the shipping lanes of Gibraltar and Mediterranean defined in Fig. 1. Monthly TROPOMI shipping $NO_2$ for 2019 and 2020 can be seen in Supplementary Figure S1(c). Figure 10(d) shows the relative change in shipping $NO_2$ from 2019 to 2020 in the Strait of Gibraltar. Using $\beta$ values and the approach described Section 2.4 and shown in Fig. 10(e), we arrive at the TROPOMI based

relative change in emission proxy shown in Fig. 10(f).

Additionally, we used AIS data to calculate an AIS based emission proxy as described in Sec. 2.5. We filtered for days with TROPOMI coverage of at least 50%. AIS data indicates that the number of ships passing per month through the Strait of Gibraltar has reduced from March 2020 onwards relative to 2019 (Fig. 10(a) and Supplementary Figure S1(a)). The average speed of the ships passing through the shipping lanes is lower between May-September 2020 compared to the same period





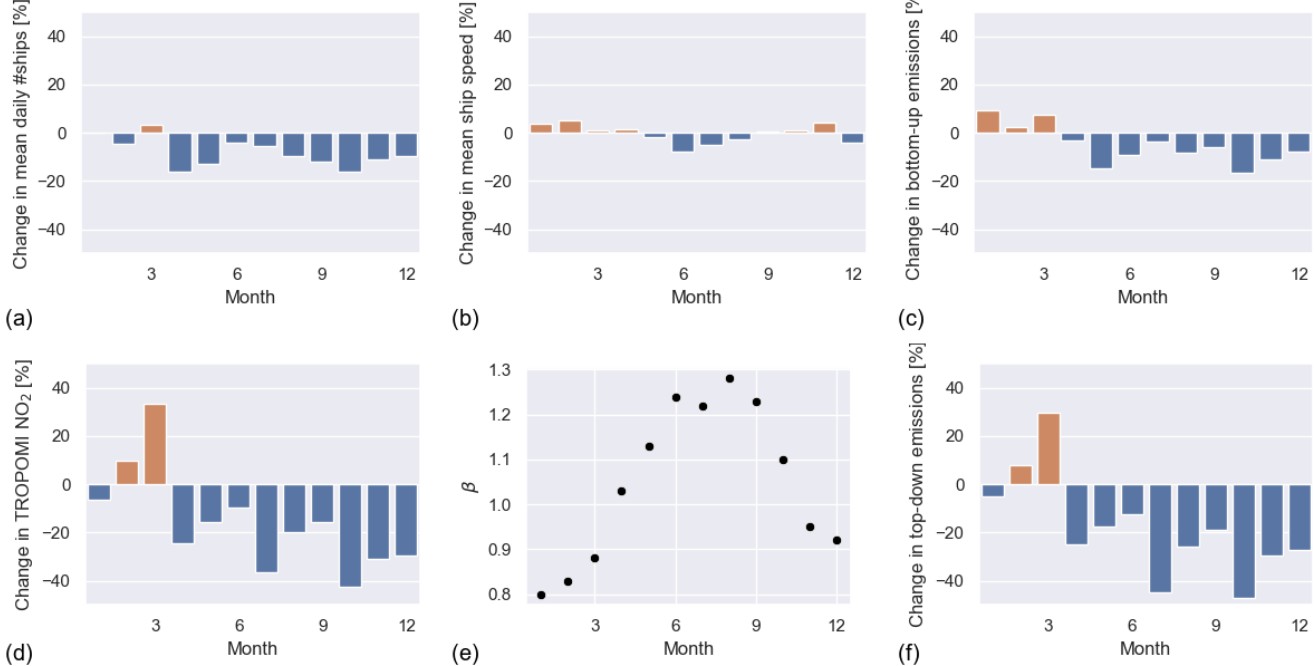

**Figure 10.** (a) Relative change in monthly mean of daily number of ships passing the Strait of Gibraltar between 2019 and 2020. (b) Same but with average ship speed. (c) Relative change in emission proxy ($v^3 \cdot L^2$). (d) Relative change in TROPOMI shipping $NO_2$. (e) Monthly $\beta$ values. (f) Relative change in $\beta \cdot NO_2$ from shipping.

in 2019 as well (Fig. 10(b) and Supplementary Figure S1(b)). This is in agreement with a study by (Millefiori et al., 2020) who found an increase in container ship speed in May and June in 2019 which is absent in 2020 leading to a relative decrease. Finally, Fig. 10(c) shows the relative change in AIS deduced emission proxy from 2019 to 2020. Similar results for the shipping lane in the Mediterranean can be found in Supplementary Figures S2 and F1.

Several studies report changes in ship activity in 2020 using AIS data. Millefiori et al. (2020) report a 5% decrease in mean
ship speed in the Mediterranean between March and April 2020 compared to 2019. The same study reported global mobility of container ships to have decreased by 10% between March and June 2020 compared to the previous year. March et al. (2021) find increases in traffic density for January and February 2020 with decreases in March-June, with Western Europe showing very strong reductions. Both studies show strong variations by vessel category and geographical distribution. Doumbia et al. (2021) find a global decrease of container ship port calls in 2020 of 7%, with a monthly local reduction of 20% for Europe in
June. The timing and magnitude of reductions reported in these studies agree with our findings: On average, AIS data indicates a reduction of 9% in April-December. The TROPOMI based emission estimates agrees in timing, but shows larger magnitude in reduction (27% in April-December).

The difference in reduction magnitude might be due to chemistry. The $\beta$ values used here are calculated on a coarse grid





(3°x2°) and might not be able to capture small-scale in-plume chemistry. Additionally, we assume the chemical conditions in
2019 and 2020 to be similar to 2006 for when the $\beta$ values were calculated. Furthermore, lateral transport complicates the
choice of $NO_2$ background as a discrimination between land and ship emissions is not possible. The $NO_2$ background in turn
has a large influence on the top-down emission estimate. Other possible sources of uncertainty lie in the different temporal and
spatial sampling of AIS and TROPOMI data and the simplified emission proxy. However, these factor are not expected to lead
to a systematic bias.

## 4    Discussion and Conclusion

We used tropospheric $NO_2$ column observations from the TROPOMI sensor to optimally monitor ship $NO_2$ pollution and study
the changes in ship $NO_x$ emissions over European seas in 2019-2020. Satellite observations of tropospheric $NO_2$ columns pro-
vide valuable information on ship air pollution over open seas, which can be used to inform compliance monitoring by flag
states and national authorities. We evaluated the high-resolution TROPOMI $NO_2$ retrievals for its potential to better detect ship
$NO_2$ pollution. In European waters alone, TROPOMI finds 6 new lanes with enhanced $NO_2$ ranging from the Aegean Sea to
the Skagerrak between Denmark and Norway, which are not detected by OMI, and which have not previously been reported in
the literature. These newly found lanes of pollution coincide with busy sailing routes and bottom-up emission proxies.

To better understand the recent detection of an individual ship's $NO_2$ plume under conditions of sun glint, we examined how
sun glint viewing geometries affect subsequent steps in the TROPOMI retrieval procedure. We find that sun glint drives higher
apparent scene reflectivity, which enhances the signal strength from spectral fitting of $NO_2$ columns along the average light
path by 20-30% over clear-sky shipping lanes. In such situations, the vertical sensitivity to $NO_2$ within the marine boundary
layer increases by up to 60%. This effect is especially strong when sea surface wind speeds are low, but non-zero. When winds
are strong, the wash causes sunlight to be reflected in other directions than directly towards the satellite, leading to little gain
in vertical sensitivity. We find that the TROPOMI $NO_2$ algorithm accounts for these effects, so that data within and outside
of sun glint geometries can be used with confidence. Nevertheless, our work clearly indicates that optimal spectral fitting can
be accomplished for small scattering angles ($<15°$) and sea surface wind speeds of 1.5-3 m/s. Although selecting a subset
fulfilling these sampling criteria reduces the amount of available data sharply, our findings indicate that sun glint conditions
are beneficial for quantifying previously undetectable small $NO_x$ emissions sources over open sea, and holding promise for
also detecting other trace gases with UV/Vis satellite instruments over water, where surface reflectivity and vertical sensitivity
is generally small.

In November 2020, KNMI implemented an improved FRESCO+ cloud retrieval called FRESCO+wide in the operational
TROPOMI $NO_2$ algorithm. We find here that this new FRESCO+wide cloud retrieval provides some 50 hPa lower cloud pres-
sures which agree better with coinciding cloud top heights from the VIIRS sensor than the standard FRESCO+. We show that
the improved cloud pressures lead to a more realistic description of vertical sensitivities in the TROPOMI $NO_2$ algorithm, and
at least partly address the known low bias in the tropospheric $NO_2$ product prior to November 2020, thus not only solving a
known issue in the TROPOMI $NO_2$ retrieval but also increasing signal strength. We then trained a neural network on a lim-





ited data set of simultaneously available standard and improved cloud and NO$_2$ retrievals. Based on 4 different training sets, the neural network learned the statistical relationship between standard FRESCO+ cloud pressures and other parameters and the new tropospheric NO$_2$ columns. We used the neural network to predict updated NO$_2$ columns for the entire 2019-2020

TROPOMI NO$_2$ record. The neural network predicts a general increase in tropospheric NO$_2$ columns. Increases are particularly strong (up to $4 \cdot 10^{15}$ molec·cm$^{-2}$) in the most polluted regions of Europe in wintertime. Our predicted (v2.1p) TROPOMI dataset enables the consistent analysis of temporal changes in NO$_2$ during the COVID-year 2020 and is useful to other data users until the TROPOMI NO$_2$ reprocessing scheduled for 2022 has been completed.

We compared changes in our v2.1p TROPOMI NO$_2$ columns between 2019 and 2020 to changes in the number of ships, their

speed and their size obtained from AIS data in the main European traffic lanes. From April 2020 onwards, TROPOMI observes 25% less NO$_2$ pollution than in the year before, in step with a 10% reduction in the number of ships and a 5% speed reduction relative to 2019. Accounting for non-linearity in local NO$_x$ chemistry, we infer an average 27% reduction in top-down NO$_x$ emissions from ships during months in which COVID-measures were in force in Europe, and global mobility decreased as a result of the pandemic. For future research, a full chemical transport modelling of AIS-based emissions and strict co-sampling

of AIS and TROPOMI data can help understanding the observed differences in top-down and bottom-up emission changes.

We showed that TROPOMI is a superior instrument to analyze relatively small enhancements in NO$_2$ pollution over dark European seas. Its vertical sensitivity to ship pollution is substantially enhanced for small scattering angles under cloud free conditions and low wind speeds. Such sun glint scenes should allow improved detection of other pollutants, such as formaldehyde and SO$_2$, as well. KNMI's operational TROPOMI NO$_2$ product is subject to continuous improvement, which causes step

changes in the publicly available data record until the official reprocessing has been finalized. Our improved (v2.1p) TROPOMI dataset offers a consistent alternative that can be used over Europe in and after 2019, and may be applied to other regions of the world where consistent NO$_2$ time series are needed.

*Data availability.* The data can be made available upon request by contacting the author (christoph.riess@wur.nl). TROPOMI L2 NO$_2$ is publicly available via the copernicus open access hub (https://scihub.copernicus.eu).





**Appendix A:  Zoomed in NO₂ maps**



**Figure A1.** Summertime mean (May-September) tropospheric NO₂ columns from TROPOMI (left panel) and summertime mean NO$_x$ emissions from the CAMS/STEAM emission inventory (right, Granier et al. (2019); Johansson et al. (2017)).





**Appendix B:** $N_{TROP,geo}$

We calculate a geometric tropospheric vertical column density $N_{trop,geo}$ using

$$N_{trop,geo} = N_{s,trop}/M_{geo} \tag{B1}$$

where $N_{s,trop}$ is the tropospheric slant column density which can be calculated from the TROPOMI files using

$\quad N_{s,trop} = N_{s,tot} - N_{s,strat} = N_{s,tot} - N_{v,strat} * M_{strat} \tag{B2}$

where $M$, $N_s$, and $N_v$ mean air mass factor, slant column density, and vertical column density, respectively. The subscripts trop, tot, and strat indicate troposhperic, total, and stratospheric columns, respectively. $M_{geo}$ can be calculated using the solar zenith angle $\theta$ and the viewing zenith angle $\theta_0$ as $M_{geo} = 1/cos(\theta) + 1/cos(\theta_0)$. The resulting tropospheric column is shown in Fig. B1.

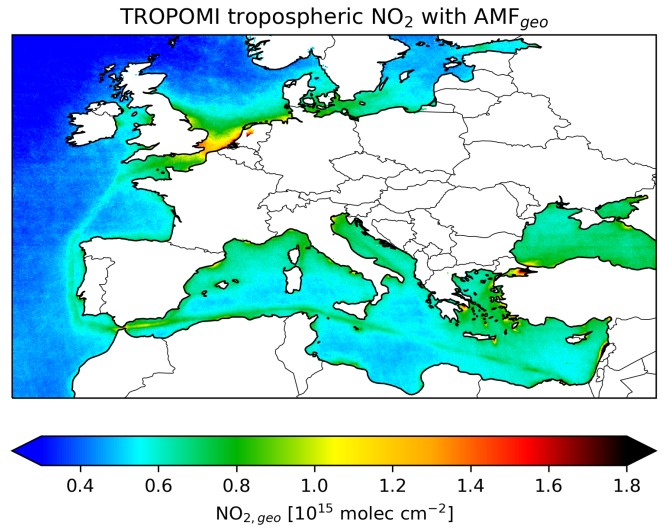

TROPOMI tropospheric $NO_2$ with $AMF_{geo}$

$NO_{2,geo}$ [$10^{15}$ molec cm$^{-2}$]

**Figure B1.** Mean of $NO_2$ columns calculated with geometrical AMF summer 2019 (MJJAS), land areas have been whitened for clarity.





## Appendix C: Spatial correlation to emissions


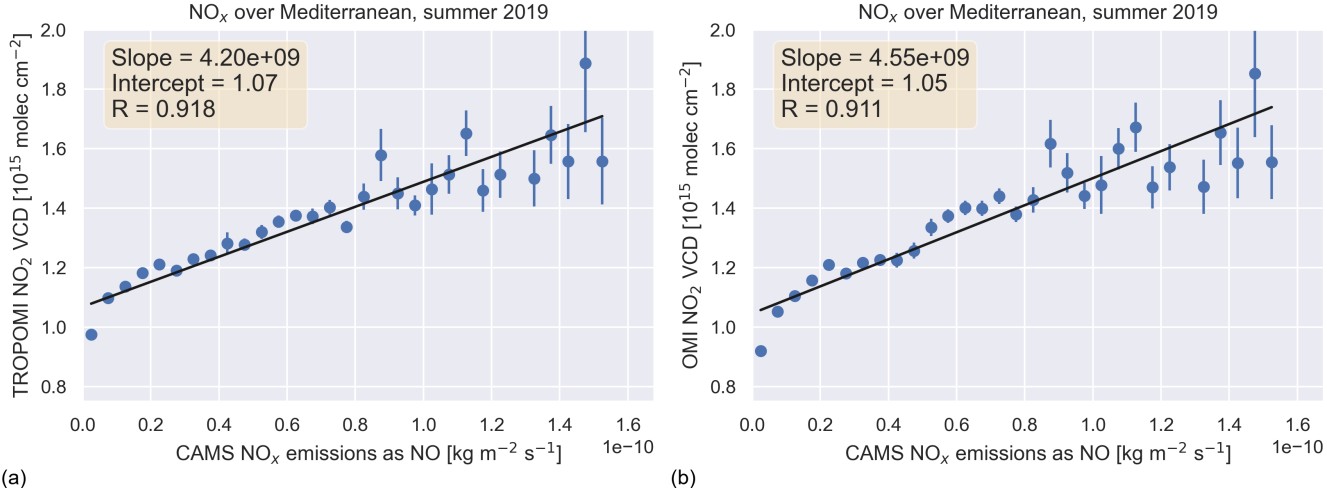

**Figure C1.** Scatter of binned summertime (May-September) 2019 tropospheric $NO_2$ columns vs emissions from CAMS/STEAM for the same period at 0.1°x0.1° in the Mediterranean. Error bars indicate the standard error of the bin. Left panel: TROPOMI, right panel: OMI.





## Appendix D: Cloud properties

### D1 Cloud fractions

The improved (v2.1) and old (v1.2) cloud fractions have strong correlation ($R^2$=0.99), but v2.1 cloud fractions are 5% lower on average, see Table D1. The spatio-temporal correlation between TROPOMI v2.1 and the well-established OMI QA4ECV cloud

fraction product is also very high ($R^2$=0.78), with TROPOMI v2.1 cloud fractions 3% lower than OMI on average. TROPOMI v2.1 shows high correlation ($R^2$=0.66) and somewhat lower cloud fractions (-11%) compared to the co-sampled effective VIIRS cloud fractions. TROPOMI cloud fractions are especially lower for partly cloud-covered scenes, possibly resulting from biased surface albedo's assumed in the TROPOMI retrieval (from the GOME-2 climatology at 0.5° resolution, see Table 1). We find similar high correlation and small differences between TROPOMI and independent data over the Mediterrenean Sea and Northwestern Europe as shown in Table D1.

**Table D1.** Evaluation of TROPOMI v2.1 cloud fractions over European shipping lanes (1-6 July 2018) against reference data.

| Shipping lane | | Mean bias | RMS | R2 | Regression |
|---|---|---|---|---|---|
| Biscay | TROPMI v2.1 vs. 1.2 | -0.020 | 0.036 | 0.99 | 0.95*x |
| | TROPOMI v2.1 vs. OMI QA4ECV | 0.002 | 0.124 | 0.78 | 0.01+0.97*x |
| | TROPOMI v2.1 vs. VIIRS | -0.062 | 0.181 | 0.66 | -0.01+0.89*x |
| Mediterranean | TROPMI v2.1 vs. 1.2 | -0.009 | 0.017 | 0.99 | 0.96*x |
| | TROPOMI v2.1 vs. OMI QA4ECV | 0.00005 | 0.09 | 0.67 | -0.01 + 1.05*x |
| | TROPOMI v2.1 vs. VIIRS | -0.050 | 0.147 | 0.60 | 0.02+0.65*x |
| NW Europe | TROPMI v2.1 vs. 1.2 | -0.015 | 0.046 | 0.95 | 0.94*x |
| | TROPOMI v2.1 vs. OMI QA4ECV | -0.038 | 0.111 | 0.76 | -0.01+0.91*x |
| | TROPOMI v2.1 vs. VIIRS | -0.026 | 0.156 | 0.64 | 0.04+0.74*x |






## D2 Cloud pressures

**Table D2.** Evaluation of TROPOMI v2.1 cloud pressures against reference data for European shipping lanes.

| Shipping lane | Product | Median cloud pressure [hPa] | 10th/90th percentile [hPa] | Geometric mean [hPa] |
|---|---|---|---|---|
| Mediterranean | TROPOMI v1.2 | 980 | 684/1010 | 920 |
| | TROPOMI v2.1 | 947 | 653/978 | 889 |
| | OMI QA4ECV | 781 | 509/903 | 739 |
| | VIIRS | 935 | 743/976 | 896 |
| NW Europe | TROPOMI v1.2 | 839 | 504/969 | 785 |
| | TROPOMI v2.1 | 861 | 590/955 | 812 |
| | OMI QA4ECV | 740 | 474/862 | 712 |
| | VIIRS | 863 | 702/993 | 853 |



## Appendix E: DNN

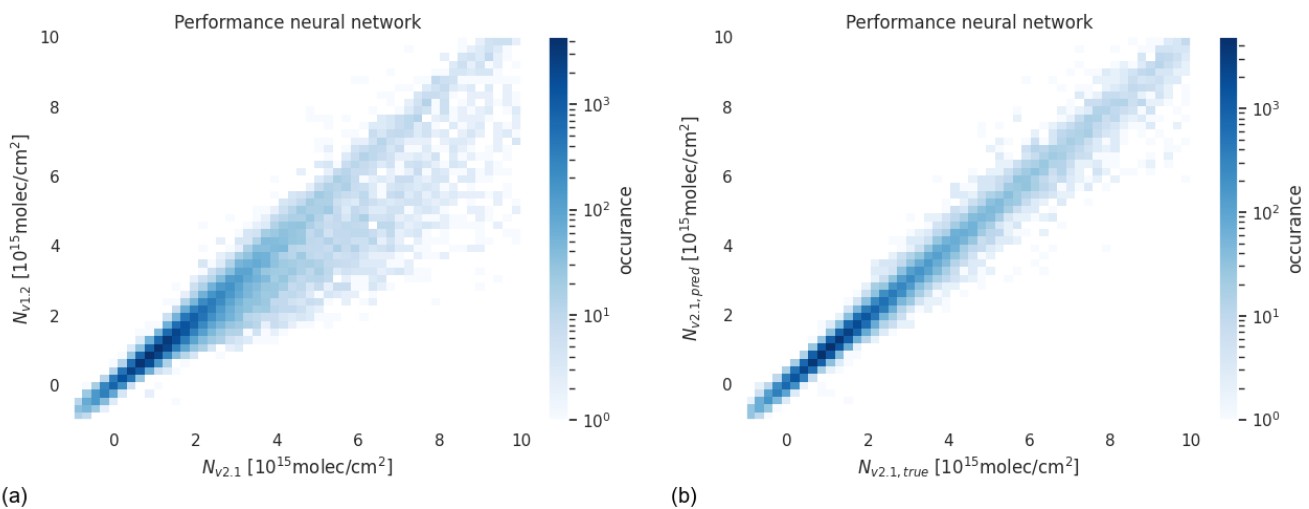

**Figure E1.** (a) Scatterplot of TROPOMI v1.2 (uncorrected) vs. actually retrieved v2.1 TROPOMI NO2 columns observed over Europe in the 4 test periods. (b) Scatterplot of DNN-predicted vs. actually retrieved v2.1 TROPOMI NO$_2$ columns observed over Europe in all areas and periods under study.

An artificial Neural Network allows us to predict v2.1 columns for the full TROPOMI mission period up to December 2020. We find the predicted v2.1 columns to be close to actual retrieved v2.1 in a testing data set. Figure E1 illustrates the skill of the DNN

approach to reliably predict v2.1 data: as a reduced major axis regression shows, the DNN-predicted v2.1 (hereafter v2.1p) NO$_2$ columns agree substantially better with the retrieved v2.1 NO$_2$ values ($N_{v2.1,true} = 0.98 \cdot N_{v2.1,pred} + 0.03 \cdot 10^{15} molec \cdot cm^{-2}$, R$^2$=0.98, n=56219) compared to the originally retrieved v1.2 NO$_2$ columns ($N_{v2.1} = 0.87 \cdot N_{v1.2} + 0.09 \cdot 10^{15} molec \cdot cm^{-2}$, R$^2 = 0.91$). The improvement from TROPOMI v1.2 to v2.1 is driven by the improved cloud pressures and associated changes in the tropospheric AMFs.

We trained the artificial Deep Neural Network (DNN) using the Python package Keras (Chollet et al., 2015) with three hidden layers. We divided the combined v1.2 and v2.1 data sets in 3 random subsets for training (60%), validation (20%), and testing (20%). The input parameters to predict TROPOMI (pseudo) v2.1 NO$_2$ columns are $N_{v,v1.2}$, $M_{trop}$, $f_{cl}$, $p_{cl}$, all viewing geometry parameters, surface albedo, and the qa value (all from v1.2). The DNN was then trained to minimize the mean absolute difference between the predicted and actually retrieved v2.1 NO$_2$ columns from the training set. This means our prediction

does not use FRESCO+wide cloud pressures for dates outside the training set period. Rather, the DNN has been trained to predict new NO$_2$ columns based on the old FRESCO+ cloud pressures and other parameters. Our DNN application succeeds in reducing the mean difference between the predicted and retrieved v2.1 NO$_2$ columns to $< 0.01 \cdot 10^{15} molec \cdot cm^{-2}$ (original v2.1 – v1.2 mean difference was $0.12 \cdot 10^{15} molec \cdot cm^{-2}$) over the 3 areas of study during the 4 periods, suggesting considerable skill in the DNN approach. Our improved data set consists of the original L2 TROPOMI NETCDF files with the predicted





change in troposhperic $NO_2$ columns as additional variable.

To show that DNN is capable of capturing seasonal variations in $NO_2$ corrections and, more broadly, that we can use a generic DNN to correct historic TROPOMI v1.2 data, we train a DNN based on 3 seasons (Summer, Winter, and Spring) and tested its predicted $NO_2$ columns against actually retrieved v2.1 data in Autumn. This analysis is done for the 3 testing areas defined in 3.3. After application of DNN, the mean discrepancy between predicted and retrieved v2.1 $NO_2$ columns reduces to

$< 0.01 \cdot 10^{15} molec \cdot cm^{-2}$ (original mean discrepancy: $0.09 \cdot 10^{15} molec \cdot cm^{-2}$) and $R^2$ improved from 0.82 to 0.97.

## Appendix F: COVID

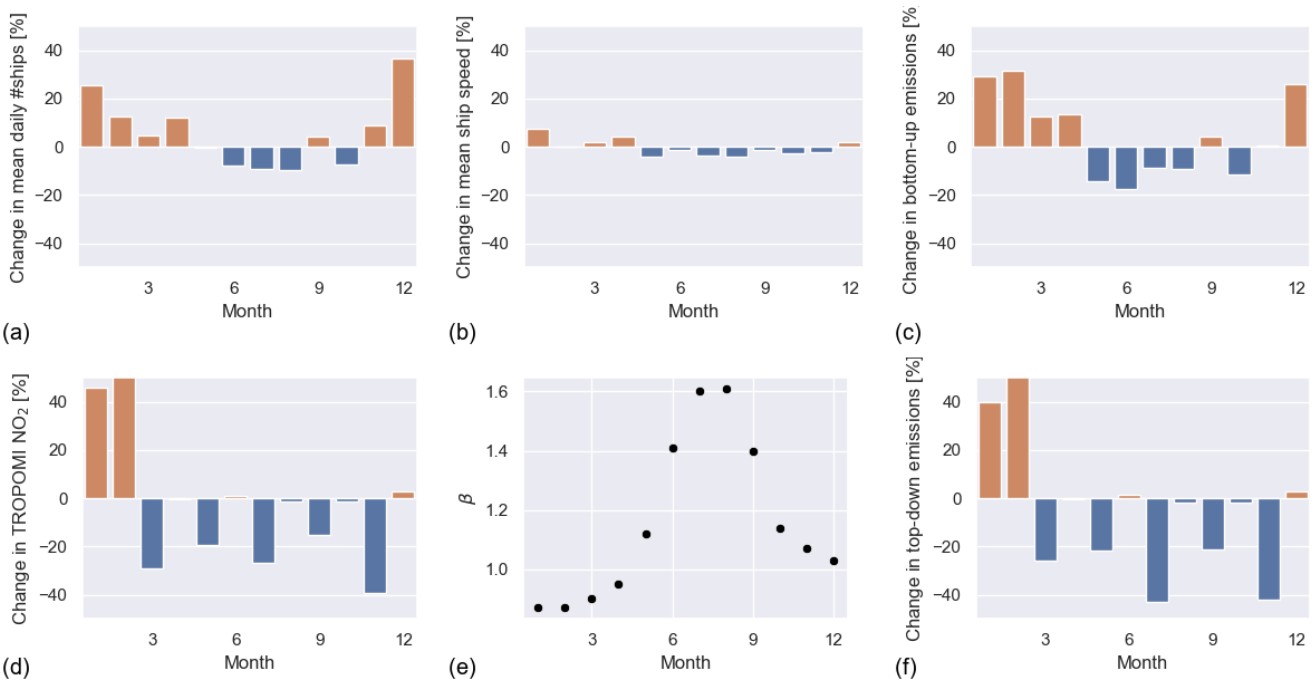

**Figure F1.** (a) Relative change in monthly mean of daily number of ships passing the Mediterranean shipping lane between 2019 and 2020. (b) Same but with average ship speed. (c) Relative change in emission proxy ($v^3 \cdot L^2$). (d) Relative change in TROPOMI shipping $NO_2$. (e) Monthly $\beta$ values. (f) Relative change in $\beta \cdot NO_2$ from shipping.



*Author contributions.* TCVWR, KFB and JvV designed the study. TCVWR performed the data analysis with support from KFB, JvV and WP. TCVWR wrote the manuscript with contributions from KFB, JvV and WP. JvV made the AIS data available. MS developed the
FRESCO+wide algorithm, JvG provided specifics on the TROPOMI $NO_2$ data versions and made the DDS-2B available. HE oversees the $NO_2$ retrieval improvements at KNMI. All authors reviewed the manuscript.

*Competing interests.* The authors declare that they have no competing interests.

*Acknowledgements.* The authors want to thank Auke van der Woude for his help regarding the Neural Network.
This work is funded by the Netherlands Human Environment and Transport Inspectorate, the Dutch Ministry of Infrastructure and Water
Management, and JvV's contribution is partly funded by the SCIPPER project, which receives funding from the European Union's Horizon 2020 research and innovation program under grant agreement Nr.814893.





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
