# Peer review of "Improved monitoring of shipping $NO_2$ with TROPOMI: decreasing $NO_x$ emissions in European seas during the COVID-19 pandemic"

_Atmospheric Measurement Techniques, 2021_

## Referee Comment (RC1)

The manuscript by Riess et al. discusses the capabilities of the TROPOMI satellite in the context of shipping emissions in Europe, the improvements of the satellite retrievals due a better cloud product, and finally the impact of the COVID lockdowns on shipping emissions. Overall, I found the manuscript to be well written with a good introduction, great instrument description and good discussion on how the FRSCO+wide cloud product improves the NO2 columns. The weakest part of this study is how the emissions changes are estimated from the NO2 columns (details can be found below). After addressing the comments listed below, I recommend the paper to be published; it would be of great interest for the readers of AMT.

**General comments**

The weakest part of this study is the "relationship between NOx emissions and columns". There are several points that are not discussed or considered here:

1) The β values are from 2006, and essentially the β determine whether or not the area is a NOx limited or saturated area, this changed very likely in some cities, not sure if shipping is areas over water would be affected by this. But this part definitely needs more discussion.
2) In the publications cited (Vinken and Verstraeten), there is the use of a γ value, which is the difference between the model and the observations over the same time period (year and month). Following the equation to look at the differences it is assumed here (but not discussed) that γ value is the same for 2019 and 2020. This might be justifiable for an entire year – where meteorology averages out. But it is an incredible simplification to assume that this would be the same for 1 month (Fig.10 and F1 show monthly changes in emissions), meteorology has a large impact on the NOx columns and it varies from year to year. This could be one of the reasons why the changes in top-down emissions are so much larger.
3) The areas chosen to estimate the emission changes: is this over water only or over shipping routes only. The relative changes of emissions is extremely influenced on the region that is chosen, including for example areas where there are no sources (background areas) tend to reduce the impact of emission change. More details on the area wold be helpful: e.g. is land excluded, is this where the majority of shipping lanes are, …
4) β value over the areas: what is the range of the β values, in Fig.10 and Fig.F1 only one single value is shown. Is this the average of the β values in that area? If so the spread should also be included – that way you will be able to add error bars in Fig.10 and F1 (e and f) that are currently missing. In Sect. 2.4 include the typical range of β, how much does this value impact the emission estimates?
5) The resolution of the β value could be an issue, the method was previously used to determine emissions over a much larger area. Is this broad resolution enough for shipping lanes and also enough for TROPOMI's relatively high resolution?

Overall, I wouldn't dispute this completely, but more discussion of the assumptions and especially weaknesses and uncertainties of using this method to determine the impact

of shipping emissions during COVID lockdowns  is necessary here. Error bars should be included in Fig. 10 and F1 (d-f).

**Specific comments**

l. 34: When have these stringent regulations been implemented?

l.48/49: This sentence sounds a little too certain about the prospects that satellites can be used for emission monitoring for ships – what are the uncertainties like? Maybe tune it down a bit – something like "..it's worth investigating if satellites can be used… "

p.5: footnote, could be included in the main text, this would flow better and the reader doesn't have to search for the footnote. It's actually important information on the difference in eh version. Also please specify the "improvements in the algorithm itself"

l.153/154: What is the relation between cloud optical thickness and cloud albedo exactly, it would be good to include the equation here.

l. 159: Is there a threshold when partly cloudy becomes "cloudy"?

l. 163: Do you have an idea how often (to what percentage) ice clouds appear. Is there maybe a different relationship that could be used for ice clouds or is this not worth it because it only happens for a small number of clouds?

l. 170: If possible, a sketch might be useful to include visualizing the relationship between the angles, if you have one handy, this would also be useful in any sort of presentations.

l. 198: $E=L^2*v^3$ is confusing, the units don't add up this would be $m^5/s^3$? Maybe better to use the term proportional some sort of factor would be missing here.

Fig.1: Include a, b, c in the panels, as done for most other figures, it's better to be consistent; also why is the inventory in the middle? To me it would be more intuitive to display the two satellite images next to each other.

Fig.2: Include label a, b, c, d

l.243: At first I was a little confused how the winds impact the albedo, but it makes sense in terms of waves (as explained a little further down), maybe put the explanation up front. Also, where does this albedo come from? From the TROPOMI files?
Fig.5: I don't quite understand the difference between a and b. what is $M_{geo}$ versus $M_{trop}$? The caption says the distribution of the tropospheric NO2 columns in both cases.

Fig.8: Is the difference v1-v2 or the other way around?

l.365/366: for which region? Gibraltar? Also discuss the results from the Mediterranean briefly.
l.369 ff: Here is some discussion on the weakness of the β value method, more needs to be were the method is described.

Supplement (2 COVID): For both Figures, the TROPOMI shipping NO2 in panel c is normalized to what? Include the area that is considered in the caption, I think it is the same as in the main manuscript l. 184, but this is not clear. The labels a, b, c should be above, for consistency.

**Technical corrections**

l. 18: "AIS" please define
p.5: footnote: "NO2" needs to be subscripted
l.141: remove "/" from VIIRS, these should simply be spaces
p.8: why a footnote? This can be just in brackets.
l.224: typo: tropospheric
l.294: molec cm$^{-2}$ shouldn't be italics, this appears a few times throughout the text, please correct all of them
p.15: why include this as a footnote? This can be put in brackets inside the text instead.
Fig.8 caption: clearness->clarity
Fig.10: other than including error bars as mentioned previously, the panel labels should be above and not below the figure as in the other figures, it's confusing to suddenly switch.
l.358: in Fig. F1 and Supplementary Fig. S2 (otherwise it sounds like F1 is in the Supplement)
l.463: molec cm$^{-2}$ shouldn't be italics
l.464: molec cm$^{-2}$ shouldn't be italics

References should be tidied up a little, many contain typos; I found the following, but there are possibly more:
l. 491 NO 2, SO 2
l. 523: incomplete
l. 540 De Ruyter de Wildt
l. 558: incomplete: journal, volume, page number missing
l. 576: Capitals?
l.639: NO 2
l. 644: Typo: Cuurent
l.645: is this the best URL?
l.653: npj Climate?
l.658: NO 2, v2 . 2
l.682 f: missing spaces
l.685: NO 2

---

## Referee Comment (RC2)

Review of the manuscript by Riess et al. "Improved monitoring of shipping NO2 with TROPOMI: decreasing NOx emissions in European seas during the COVID-19 pandemic"

The manuscript presents an analysis of ship emissions over European seas based on TROPOMI NO2 observations. The authors also analyse the effects of COVID-19 restrictions on shipping and the relative decrease in emissions. The manuscript has sufficient elements of novelties as it provides a deeper analysis of the capabilities of TROPOMI NO2 observations for ship emission monitoring, after a first paper dedicated to the detection of individual plumes by Georgoulias et al. (2020). I recommend publication after addressing the following comments:

1. You mention the NO2 profiles used in NO2 retrieval from TM5 at 1x1degrees: Can you comment on the possible uncertainty related to such coarse resolution and on their accuracy over sea in particular?
2. Did you assess how the FRESCO+wide perform over ice/snow surfaces? Can you comment on that?
3. How do your simpler emission estimates from AIS compare to the estimates from STEAM model? And why don't you use STEAM emissions for the analysis of changes? Not available for 2020? Also, you use this CAMS-STEAM emission data in Fig. 1 and 2, maybe you should introduce this dataset a little bit earlier.
4. L340-… This statement is not supported, are you implicitly referring to your figure 10c? if yes, please make that connection.
5. Can you address and discuss a bit more the uncertainties on these monthly $\beta$ values at such coarse resolution when you use it here for the emission change estimates? Also, how could monthly $\beta$ values change between 2006 and 2020 due to meteorology or other factors?

Technical comments:

L128-L129 "in order to distinguish between bright reflecting layers at the Earth's surface from reflecting surfaces in the lower atmosphere.": remove "between" or replace "from" with "and"

L359-360 you just said this in the previous paragraph, maybe rephrase here

---

## Author Comment (AC1)

**RC#1**

The manuscript by Riess et al. discusses the capabilities of the TROPOMI satellite in the context of shipping emissions in Europe, the improvements of the satellite retrievals due a better cloud product, and finally the impact of the COVID lockdowns on shipping emissions. Overall, I found the manuscript to be well written with a good introduction, great instrument description and good discussion on how the FRSCO+wide cloud product improves the NO2 columns. The weakest part of this study is how the emissions changes are estimated from the NO2 columns (details can be found below). After addressing the comments listed below, I recommend the paper to be published; it would be of great interest for the readers of AMT.

We thank referee #1 for his suggestions and encouraging words. For detailed replies to the comment see below.

**General comments**

The weakest part of this study is the "relationship between NOx emissions and columns". There are several points that are not discussed or considered here:

1) The β values are from 2006, and essentially the β determine whether or not the area is a NOx limited or saturated area, this changed very likely in some cities, not sure if shipping is areas over water would be affected by this. But this part definitely needs more discussion.

We agree. In the manuscript we have revised our approach to account for the non-linear relationship between $NO_x$ emissions and $NO_2$ columns. We now use the higher resolution $β$ from Vinken et al. [2014] calculated for 2006 and include a discussion (in section 3.5 + Supplement 4) on variability in $β$ from year to year, and how this could have influenced the top-down $NO_x$ emission estimates.

To evaluate possible changes in chemical regime between 2019 and 2020, we studied the EAC4 re-analysis of atmospheric composition (Inness et al., 2021). $NO_2$ and $O_3$ monthly mean columns show only modest year-to-year variability between 2006 and 2019, as shown in Figure 1 below. This suggests modest variability in the chemical regimes (and therefore $β$) between 2006 and 2019.

[Figure]

Figure 1. Monthly mean (12UTC) $NO_2$ (top) and $O_3$ (bottom) tropospheric columns for Gibraltar (left, 5°W-0°E, 35°N-37°N) and Eastern Mediterranean (right, 32°N-36°N, 15°E-20°E) from EAC4.

2) In the publications cited (Vinken and Verstraeten), there is the use of a γ value, which is the difference between the model and the observations over the same time period (year and month). Following the equation to look at the differences it is assumed here (but not discussed) that γ value is the same for 2019 and 2020. This might be justifiable for an entire year – where meteorology averages out. But it is an incredible simplification to assume that this would be the same for 1 month (Fig.10 and F1 show monthly changes in emissions), meteorology has a large impact on the NOx columns and it varies from year to year. This could be one of the reasons why the changes in top-down emissions are so much larger.

By definition, the γ value accounts for for updates to the a priori $NO_2$ profile shape due to $NO_x$ emission changes (see Vinken et al., 2014). However, the TROPOMI $NO_2$ retrievals used here do not account for changes in the a priori $NO_2$ profiles resulting from sudden emission changes. Since we infer here only modest emission changes between 2019 to 2020, so that the uncertainty in a priori profiles caused by the $1° \times 1°$ TM5 resolution is the dominant source of uncertainty, this is well justified.

Because differences in meteorology between months in 2019 compared to 2020 could be causing differences in corresponding $NO_2$ columns (see Figure 1 above), we should indeed be careful to not interpreted these as being caused by $NO_x$ emission differences.

To assess the possible influence of meteorological differences on $NO_2$ columns, we compared CAMS European Air Quality Forecast monthly mean data for the same months in 2019 and 2020 (METEO FRANCE et al., 2020). These simulations take differences in meteorology into account but keep $NO_x$ emissions constant for 2019 and 2020. The monthly mean $NO_2$ columns for 12UTC are shown in Figure 2 below.

Over Gibraltar, the CAMS model ensemble mean predicts 5%-20% reductions in $NO_2$ columns for January to November 2020 relative to January-November 2019.  For the Eastern Mediterranean the CAMS ensemble predicts lower $NO_2$ columns for all months in 2020 compared to 2019 with the exception of July and November.

The CAMS forecasts therefore suggest that the observed changes in TROPOMI shipping $NO_2$ are caused by both emission changes as well as meteorological differences, since meteorology alone cannot explain the observed increases in January-March nor the reductions in April-December. In the revised manuscript we use the changes in predicted $NO_2$ columns to estimate the uncertainty imposed by meteorological variability on the inferred $NO_x$ emissions.

[Figure]

Figure 2. Monthly NO$_2$ column for 12UTC CAMS ensemble mean forecast in the Strait of (left, 5°W-0°E, 35°N-37°N) and Eastern Mediterranean (right, 32°N-36°N,15°E-30°E) in 2019 (green) and 2020 (red).

3) The areas chosen to estimate the emission changes: is this over water only or over shipping routes only. The relative changes of emissions is extremely influenced on the region that is chosen, including for example areas where there are no sources (background areas) tend to reduce the impact of emission change. More details on the area wold be helpful: e.g. is land excluded, is this where the majority of shipping lanes are, …

We use areas over water only as indicated in Fig1(c) by the purple rectangles, and discussed in Section 3.4. We clarified this in the corresponding lines in Section 3.4 and 3.5. These areas are among the most frequently travelled shipping lanes around Europe as indicated by CAMS/STEAM emissions in Fig1(c) and AIS data. Our top-down NO$_x$ emission estimates are relatively insensitive to the selection of the area. To corroborate this further, we performed a sensitivity analysis with a 10% more narrow area in lateral direction which yielded similar (less than 5% different) results.

4) β value over the areas: what is the range of the β values, in Fig.10 and Fig.F1 only one single value is shown. Is this the average of the β values in that area? If so the spread should also be included – that way you will be able to add error bars in Fig.10 and F1 (e and f) that are currently missing. In Sect. 2.4 include the typical range of β, how much does this value impact the emission estimates?

Thank you for bringing this up. We indeed take the average of the β values in the areas defined in Section 2.4. We now include the standard deviation of β values in an area as indicator for the uncertainty in β. The uncertainty ranges from 0.05-0.13 for Gibraltar. Uncertainty intervals are now included in the Fig. 10 (see also below).

5) The resolution of the β value could be an issue, the method was previously used to determine emissions over a much larger area. Is this broad resolution enough for shipping lanes and also enough for TROPOMI's relatively high resolution?

This is a good point, which made us reconsider and revise our choice of β values. We now replace the β values from Verstraeten et al. [2015] by those calculated specifically

for shipping lanes as in Vinken et al. [2014]. The latter have the advantage of being simulated at a higher spatial resolution of $50 \times 70$ km$^2$, and have been calculated with a plume-in-grid ship emission parameterization, which is more representative for monthly averaged NO$_2$ signals over shipping lanes than from Verstraeten et al. [2015] where ship emissions are instantaneously diluted over $200 \times 300$ km$^2$ grid cells. The $\beta$ values of Vinken show a similar seasonality as those of Verstraeten et al. [2015], but are lower by ±25%, which better reflects the polluted character of our selected areas, as discussed in detail in Vinken et al. [2014], section 3.2.

The NO$_2$ signals studied here represent changes in mean shipping lane NO$_2$ columns and not individual plumes. Figure 1(c) and Figure 9 also clearly show that the areas where NO$_2$ columns show changes have a width in the order of 100km, which is comparable to the resolution of the $\beta$ values from Vinken et al. [2014]. The resolution of $\beta$ values from Vinken et al. [2014] can thus be considered appropriate for our purpose.

Overall, I wouldn't dispute this completely, but more discussion of the assumptions and especially weaknesses and uncertainties of using this method to determine the impact of shipping emissions during COVID lockdowns is necessary here. Error bars should be included in Fig. 10 and F1 (d-f).

We now include a discussion on the assumptions and uncertainties associated with the top-down NO$_x$ emission estimates in Section 3.5, as suggested by the referee. We estimate the uncertainty of the top-down emission changes to be driven by uncertainties in the $\beta$ values, from assumption on meteorological representativeness and from the area selected to be:

$$dE^2 = (\sigma_{area} * \beta)^2 + (\sigma_{meteo} * \beta)^2 + (dN * \sigma_\beta)^2$$

Where $dN$ is the relative change in TROPOMI NO$_2$, $\sigma_{area}$= 5% as the sensitivity of the TROPOMI NO$_2$ to the area of study, $\sigma_{meteo}$= 16% for Gibraltar and 11% for the Eastern Mediterranean the impact of meteorology (and therefore transport & lifetime changes) on column changes, $\sigma_\beta$ the combined spatial and temporal variability of $\beta$ in the area of study estimated to be 0.15 (dimensionless) from the spatial spread and the year-to-year variability in the $\beta$ values. These uncertainties are now included in the revised manuscript and shown as error bars in Figure 10(d-f) and F1(d-f).

While single TROPOMI NO$_2$ columns have substantial (random and systematic) uncertainties, these largely cancel out when taking the relative differences between months in different years. Averaging over space and over a month smoothes out the random error while the systematic errors cancel out largely in the relative changes studied here. This renders the uncertainty introduced by the satellite measurements to be negligible in our estimates of emission changes between 2019 and 2020.

***Specific comments***

l. 34: When have these stringent regulations been implemented?
There are different regulations for $NO_x$ and sulfur emissions, each further divided in different Tiers. In short, Tier II restrictions apply for ships build in 2011 or later, Tier III restrictions apply for new-built ships operating in North Sea and Baltic starting 2021. Examples are given in lines 35-39 in the introduction and a complete overview is given in Supplemental Material section 1.

l.48/49: This sentence sounds a little too certain about the prospects that satellites can be used for emission monitoring for ships – what are the uncertainties like? Maybe tune it down a bit – something like "..it's worth investigating if satellites can be used…"
Given the Georgoulias et al. [2020] paper we were optimistic here. However, we will rephrase it to "satellite remote sensing offers a promising alternative".

p.5: footnote, could be included in the main text, this would flow better and the reader doesn't have to search for the footnote. It's actually important information on the difference in eh version. Also please specify the "improvements in the algorithm itself"
Thank you for the remark, we have changed this accordingly.

l.153/154: What is the relation between cloud optical thickness and cloud albedo exactly, it would be good to include the equation here.
In general terms, the cloud albedo increases with cloud optical thickness for water clouds. The non-linear relationship between cloud optical thickness and cloud albedo has been taken from Buriez et al. [2005]. The 6th order polynomial relationship is now included in a footnote in Section 2.2.

l. 159: Is there a threshold when partly cloudy becomes "cloudy"?
In the TROPOMI $NO_2$-retrieval, partly cloudy becomes cloudy when the cloud radiance fraction exceeds 0.5. This usually occurs (depending on the surface albedo) for cloud fractions larger than 0.2. For the cloud fraction comparison we evaluate all data with *cf* >0.05 (and *qa*>0.5). For the study of cloud pressures (3.3.1) only scenes with 0.05<*cf*<0.2 were used.

l. 163: Do you have an idea how often (to what percentage) ice clouds appear. Is there maybe a different relationship that could be used for ice clouds or is this not worth it because it only happens for a small number of clouds?
Around 25-30% of VIIRS cloud retrievals in the area of study are ice water clouds. But ice water clouds are located in the higher atmosphere only (typically 310hPa vs 760hPa for liquid water clouds for January 2019, 280hPa vs 760hPa in June 2018). Here, improved cloud pressures have only little influence on $NO_2$ columns as $NO_2$ pollution in the upper troposphere is usually very small (see Fig.7). A clarifying sentence has been added to Section 3.3.2.

l. 170: If possible, a sketch might be useful to include visualizing the relationship between Skethe angles, if you have one handy, this would also be useful in any sort of presentations.
The sketch shown below has been included in the supplement for clarity.

[Figure]

*Figure 3 Sketch of satellite and solar geometry. The thick and thin yellow lines indicate the incoming and directly reflected solar light, respectively. The blue arrow indicates the light scattered in the direction of the satellite. The large blue arc is the Earth's surface. In green we show the difference of the viewing (satellite) and solar azimuth angle, orange and purple show the viewing and solar zenith angles, respectively. The thick black arc is the scattering angle, i.e. the angle between the reflected sunlight and the viewing direction.*

l. 198: E=L2*v3 is confusing, the units don't add up this would be m5/s3? Maybe better to use the term proportional some sort of factor would be missing here.
This is indeed a proportionality and has been adopted in the revised manuscript. As we use it to study relative changes, the proportionality factor cancels out.

Fig.1: Include a, b, c in the panels, as done for most other figures, it's better to be consistent; also why is the inventory in the middle? To me it would be more intuitive to display the two satellite images next to each other.
We adapted this in the revised manuscript., the Figure now includes labels and the two satellite images are next to each other.

Fig.2: Include label a, b, c, d
This has been adopted in the revised manuscript.

l.243: At first I was a little confused how the winds impact the albedo, but it makes sense in terms of waves (as explained a little further down), maybe put the explanation up front. Also, where does this albedo come from? From the TROPOMI files?
The explanation with waves is part of the same paragraph (but two pages further down), we adopted the first sentence of the paragraph to "… by focusing on scenes with low-moderate wind speeds (2 m/s) : as wind-induced waves are expected to change the reflectivity" to make it clear.
Yes, the scene albedo comes from the TROPOMI file.

Fig.5: I don't quite understand the difference between a and b. what is Mgeo versus Mtrop? The caption says the distribution of the tropospheric NO2 columns in both cases.

*$M_{geo}$ refers to the geometrical AMF, which accounts for the viewing geometry but neglects the reductions in vertical sensitivity to $NO_2$. $M_{trop}$ accounts for both viewing geometry and the vertical sensitivity to $NO_2$. $N/M_{geo}$ is therefore the slant column normalized for the viewing geometry only. $M_{trop}$ is used to calculate the actual VCD in the official TROPOMI product. $M_{geo}$ is introduced in Appendix B. Fig. 5 basically shows the slant column corrected for the viewing geometry, suggesting enhanced $NO_2$ detected in the slant column. This enhanced $NO_2$ vanishes when accounting for the enhanced surface reflectivity as is done correctly by the TROPOMI algorithm and shown in Fig 5b.*

Fig.8: Is the difference v1-v2 or the other way around?
*This is new-old (so v2.1p-v1.2/v1.3). We added it to the figure caption in the revised manuscript.*

l.365/366: for which region? Gibraltar? Also discuss the results from the Mediterranean briefly.
*Yes, this is for Gibraltar (Fig 10). A discussion of the Mediterranean has been added to the revised manuscript.*

l.369 ff: Here is some discussion on the weakness of the β value method, more needs to be were the method is described.
*We have revised our approach and discuss in more detail the assumptions and uncertainties associated with our selection of the β value in Section 3.5.*

Supplement (2 COVID): For both Figures, the TROPOMI shipping NO2 in panel c is normalized to what? Include the area that is considered in the caption, I think it is the same as in the main manuscript l. 184, but this is not clear. The labels a, b, c should be above, for consistency.
*This is normalized to the maximum monthly value. Yes, this is the same area as before. Figure label positions are consistent with the main text.*

**Technical corrections**

l. 18: "AIS" please define *A definition has been added here.*
p.5: footnote: "NO2" needs to be subscripted *This has been corrected.*
l.141: remove "/" from VIIRS, these should simply be spaces *This has been corrected.*
p.8: why a footnote? This can be just in brackets. *This has been adjusted.*
l.224: typo: tropospheric *This has been corrected.*
l.294: molec cm-2 shouldn't be italics, this appears a few times throughout the text, please correct all of them *This has been corrected throughout the manuscript.*
p.15: why include this as a footnote? This can be put in brackets inside the text instead. *This has been changed.*
Fig.8 caption: clearness->clarity *This has been changed.*

Fig.10: other than including error bars as mentioned previously, the panel labels should be above and not below the figure as in the other figures, it's confusing to suddenly switch.
The panel labels in all figures are consistently below the subfigure
l.358: in Fig. F1 and Supplementary Fig. S2 (otherwise it sounds like F1 is in the Supplement) This has been changed.
l.463: molec cm-2 shouldn't be italics This has been corrected.
l.464: molec cm-2 shouldn't be italics This has been corrected.

References should be tidied up a little, many contain typos; I found the following, but there are possibly more: Several further references have been tidied up.
l. 491 $NO_2$, $SO_2$ This has been corrected.
l. 523: incomplete This reference is correct.
l. 540 De Ruyter de Wildt This has been corrected.
l. 558: incomplete: journal, volume, page number missing This has been completed.
l. 576: Capitals? This has been corrected.
l.639: $NO_2$ This has been corrected.
l. 644: Typo: Cuurent This has been corrected.
l.645: is this the best URL? This has been changed.
l.653: npj Climate? This reference is correct.
l.658: $NO_2$, v2 . 2 This has been corrected.
l.682 f: missing spaces This has been corrected.
l.685: NO 2 This has been corrected.

**References**

Inness, A., Ades, M., Agustí-Panareda, A., Barré, J., Benedictow, A., Blechschmidt, A., Dominguez, J., Engelen, R., Eskes, H., Flemming, J., Huijnen, V., Jones, L., Kipling, Z., Massart, S., Parrington, M., Peuch, V.-H., Razinger M., Remy, S., Schulz, M., & Suttie, M. (2021). *CAMS global reanalysis (EAC4) monthly averaged fields*. https://ads.atmosphere.copernicus.eu/cdsapp#!/dataset/cams-global-reanalysis-eac4-monthly?tab=overview

METEO FRANCE, Institut national de l'environnement industriel et des risques (Ineris), Aarhus University, Norwegian Meteorological Institute (MET Norway), Jülich Institut für Energie- und Klimaforschung (IEK), Institute of Environmental Protection – National Research Institute (IEP-NRI), Koninklijk Nederlands Meteorologisch Instituut (KNMI), Nederlandse Organisatie voor toegepast-natuurwetenschappelijk onderzoek (TNO), Swedish Meteorological and Hydrological Institute (SMHI), & Finnish Meteorological Institute (FMI). (2020). *CAMS European air quality forecasts*. https://ads.atmosphere.copernicus.eu/cdsapp#!/dataset/cams-europe-air-quality-forecasts?tab=overview

Verstraeten, W. W., Neu, J. L., Williams, J. E., Bowman, K. W., Worden, J. R., & Boersma, K. F. (2015). Rapid increases in tropospheric ozone production and export from China. *Nature Geoscience, 8*(9), 690–695. https://doi.org/10.1038/ngeo2493

Vinken, G. C. M., Boersma, K. F., van Donkelaar, A., & Zhang, L. (2014). Constraints on ship NOx emissions in Europe using GEOS-Chem and OMI satellite NO2 observations. *Atmospheric Chemistry and Physics, 14*(3), 1353–1369. https://doi.org/10.5194/acp-14-1353-2014

---

## Author Comment (AC2)

**RC#2**

Review of the manuscript by Riess et al. "Improved monitoring of shipping NO2 with TROPOMI: decreasing NOx emissions in European seas during the COVID-19 pandemic"

The manuscript presents an analysis of ship emissions over European seas based on TROPOMI NO2 observations. The authors also analyse the effects of COVID-19 restrictions on shipping and the relative decrease in emissions. The manuscript has sufficient elements of novelties as it provides a deeper analysis of the capabilities of TROPOMI NO2 observations for ship emission monitoring, after a first paper dedicated to the detection of individual plumes by Georgoulias et al. (2020). I recommend publication after addressing the following comments:

We want to thank reviewer #2 for their comments. Please see below for replies to the specific comments

1. You mention the NO2 profiles used in NO2 retrieval from TM5 at 1x1degrees: Can you comment on the possible uncertainty related to such coarse resolution and on their accuracy over sea in particular?

The reviewer brings up an important point. Coarse a-priori profiles likely underestimate the $NO_2$ concentrations in the boundary layer over busy shipping lanes and overestimate $NO_2$ concentrations outside of the shipping lanes, as the emission spatial patterns are not resolved by the model grid. This issue for TROPOMI $NO_2$ retrievals is in line with earlier findings for other satellite instruments: coarse-gridded TM3 profiles used for GOME were estimated to give rise to an error of 10% on the retrieved columns (Boersma et al., 2004). Heckel et al. [2011] found that coarse resolution a-priori profiles can cause errors larger than $2*10^{15}$ molec/cm$^2$ for individual pixels for OMI. This error has a systematic component and can therefore not be averaged out, further highlighting the importance of high-resolution a-priori information.
We are currently analyzing low-altitude aircraft measurements over the North Sea to improve our understanding of $NO_2$ vertical profiles over sea, and the capability of TM5 to simulate these.

2. Did you assess how the FRESCO+wide perform over ice/snow surfaces? Can you comment on that?

No, we did not assess this as ice conditions are uncommon in the study areas of this paper.

3. How do your simpler emission estimates from AIS compare to the estimates from STEAM model? And why don't you use STEAM emissions for the analysis of changes? Not available for 2020? Also, you use this CAMS-STEAM emission data in Fig. 1 and 2, maybe you should introduce this dataset a little bit earlier.

STEAM emissions for 2020 are not available yet to our knowledge. We have not attempted a comparison on a single-ship basis either as the data were not available to us.
We now introduce CAMS-STEAM data earlier in the same paragraph.

4. L340-… This statement is not supported, are you implicitly referring to your figure 10c? if yes, please make that connection.

A reference to Figure 10c has been added in the revised manuscript.

5. Can you address and discuss a bit more the uncertainties on these monthly $\beta$ values at such coarse resolution when you use it here for the emission change estimates? Also, how could monthly $\beta$ values change between 2006 and 2020 due to meteorology or other factors?

This is a good point, which made us reconsider and revise our choice of $\beta$ values. We now replace the $\beta$ values from Verstraeten et al. [2015] by those calculated specifically for shipping lanes as in Vinken et al. [2014]. The latter have the advantage of being simulated at a higher spatial resolution of $50\times70$ km$^2$, and have been calculated with a plume-in-grid ship emission parameterization, which is more representative for monthly

averaged NO₂ signals over shipping lanes than from Verstraeten et al. [2015] where ship emissions are instantaneously diluted over 200×300 km² grid cells. The *β* values of Vinken show a similar seasonality as those of Verstraeten et al. [2015], but are lower by ±25%, which better reflects the polluted character of our selected areas, as discussed in detail in Vinken et al. [2014], section 3.2.

The NO₂ signals studied here represent changes in mean shipping lane NO₂ columns and not individual plumes. Figure 1(c) and Figure 9 also clearly show that the areas where NO₂ columns show changes have a width in the order of 100km, which is comparable to the resolution of the *β* values from Vinken et al. [2014]. The resolution of *β* values from Vinken et al. [2014] can thus be considered appropriate for our purpose.

[Figure]

Figure 1. Monthly mean (12UTC) NO₂ (top) and O₃ (bottom) tropospheric columns for Gibraltar (left, 5°W-0°E, 35°N-37°N) and Eastern Mediterranean (right, 32°N-36°N,15°E-20°E) from EAC4.

To evaluate possible changes in chemical regime between 2019 and 2020, we studied the EAC4 re-analysis of atmospheric composition (Inness et al., 2021). NO₂ and O₃ monthly mean columns show only modest year-to-year variability between 2006 and 2019, as shown in Figure 1. This suggests modest variability in the chemical regimes (and therefore *β*) between 2006 and 2019.

To assess the possible influence of meteorological differences on NO₂ columns, we compared CAMS European Air Quality Forecast data for the same months in 2019 and

2020 (METEO FRANCE et al., 2020). These simulations take differences in meteorology into account but keep $NO_x$ emissions constant for 2019 and 2020. The monthly mean $NO_2$ columns for 12UTC are shown in Figure 2.

[Figure]

Figure 2. Monthly $NO_2$ column for 12UTC CAMS ensemble mean forecast in the Strait of (left, 5°W-0°E, 35°N-37°N) and Eastern Mediterranean (right, 32°N-36°N,15°E-30°E) in 2019 (green) and 2020 (red).

Over Gibraltar, the CAMS model ensemble mean predicts 5%-20% reductions in $NO_2$ columns for January to November 2020 relative to January-November 2019. For the Eastern Mediterranean the CAMS ensemble predicts lower $NO_2$ columns for all months in 2020 compared to 2019 with the exception of July and November.

The CAMS forecasts therefore suggest that the observed changes in TROPOMI shipping $NO_2$ are caused by both emission changes as well as meteorological differences, since meteorology alone cannot explain the observed increases in January-March nor the reductions in April-December. In the revised manuscript we use the changes in predicted $NO_2$ columns to estimate the uncertainty imposed by meteorological variability on the inferred $NO_x$ emissions.

We now include a discussion on the assumptions and uncertainties associated with the top-down $NO_x$ emission estimates in Section 3.5, as suggested by the referee. We estimate the uncertainty of the top-down emission changes to be driven by uncertainties in the $\beta$ values, from assumption on meteorological representativeness and from the area selected to be:

$$dE^2 = (\sigma_{area} * \beta)^2 + (\sigma_{meteo} * \beta)^2 + (dN * \sigma_\beta)^2$$

Where $dN$ is the relative change in TROPOMI $NO_2$, $\sigma_{area}$= 5% as the sensitivity of the TROPOMI $NO_2$ to the area of study, $\sigma_{meteo}$= 16% for Gibraltar and 11% for the Eastern Mediterranean the impact of meteorology (and therefore transport & lifetime changes) on column changes, $\sigma_\beta$ the combined spatial and temporal variability of $\beta$ in the area of study estimated to be 0.15 (dimensionless) from the spatial spread and the year-to-year variability in the $\beta$ values. These uncertainties are now included in the revised manuscript and shown as error bars in Figure 10(d-f) and F1(d-f).

While single TROPOMI $NO_2$ columns have substantial (random and systematic) uncertainties, these largely cancel out when taking the relative differences between months in different years. Averaging over space and over a month smoothes out the random error while the systematic errors cancel out largely in the relative changes

studied here. This renders the uncertainty introduced by the satellite measurements to be negligible in our estimates of emission changes between 2019 and 2020.

**Technical comments:**
L128-L129 "in order to distinguish between bright reflecting layers at the Earth's surface
from reflecting surfaces in the lower atmosphere.": remove "between" or replace "from" with "and"
"From" has been replaced with "and".
L359-360 you just said this in the previous paragraph, maybe rephrase here
This has been rephrased.

**References**

Boersma, K. F., Eskes, H. J., & Brinksma, E. J. (2004). Error analysis for tropospheric NO2 retrieval from space. *Journal of Geophysical Research D: Atmospheres*, *109*(4). https://doi.org/10.1029/2003jd003962

Heckel, A., Kim, S.-W., Frost, G. J., Richter, A., Trainer, M., & Burrows, J. P. (2011). Atmospheric Measurement Techniques Influence of low spatial resolution a priori data on tropospheric NO2 satellite retrievals. *Atmos. Meas. Tech*, *4*, 1805–1820. https://doi.org/10.5194/amt-4-1805-2011

Inness, A., Ades, M., Agustí-Panareda, A., Barré, J., Benedictow, A., Blechschmidt, A., Dominguez, J., Engelen, R., Eskes, H., Flemming, J., Huijnen, V., Jones, L., Kipling, Z., Massart, S., Parrington, M., Peuch, V.-H., Razinger M., Remy, S., Schulz, M., & Suttie, M. (2021). *CAMS global reanalysis (EAC4) monthly averaged fields*. https://ads.atmosphere.copernicus.eu/cdsapp#!/dataset/cams-global-reanalysis-eac4-monthly?tab=overview

METEO FRANCE, Institut national de l'environnement industriel et des risques (Ineris), Aarhus University, Norwegian Meteorological Institute (MET Norway), Jülich Institut für Energie- und Klimaforschung (IEK), Institute of Environmental Protection – National Research Institute (IEP-NRI), Koninklijk Nederlands Meteorologisch Instituut (KNMI), Nederlandse Organisatie voor toegepast-natuurwetenschappelijk onderzoek (TNO), Swedish Meteorological and Hydrological Institute (SMHI), & Finnish Meteorological Institute (FMI). (2020). *CAMS European air quality forecasts*. https://ads.atmosphere.copernicus.eu/cdsapp#!/dataset/cams-europe-air-quality-forecasts?tab=overview

Verstraeten, W. W., Neu, J. L., Williams, J. E., Bowman, K. W., Worden, J. R., & Boersma, K. F. (2015). Rapid increases in tropospheric ozone production and export from China. *Nature Geoscience, 8*(9), 690–695. https://doi.org/10.1038/ngeo2493

Vinken, G. C. M., Boersma, K. F., van Donkelaar, A., & Zhang, L. (2014). Constraints on ship NOx emissions in Europe using GEOS-Chem and OMI satellite NO2 observations. *Atmospheric Chemistry and Physics*, *14*(3), 1353–1369. https://doi.org/10.5194/acp-14-1353-2014

---

## Author Response (AR1)

**RC#1**

The manuscript by Riess et al. discusses the capabilities of the TROPOMI satellite in the context of shipping emissions in Europe, the improvements of the satellite retrievals due a better cloud product, and finally the impact of the COVID lockdowns on shipping emissions. Overall, I found the manuscript to be well written with a good introduction, great instrument description and good discussion on how the FRSCO+wide cloud product improves the NO2 columns. The weakest part of this study is how the emissions changes are estimated from the NO2 columns (details can be found below). After addressing the comments listed below, I recommend the paper to be published; it would be of great interest for the readers of AMT.

We thank referee #1 for his suggestions and encouraging words. For detailed replies to the comment see below.

**General comments**

The weakest part of this study is the "relationship between NOx emissions and columns". There are several points that are not discussed or considered here:

 The β values are from 2006, and essentially the β determine whether or not the area is a NOx limited or saturated area, this changed very likely in some cities, not sure if shipping is areas over water would be affected by this. But this part definitely needs more discussion.

We agree. In the manuscript we have revised our approach to account for the nonlinear relationship between NOx emissions and NO2 columns. We now use the higher resolution  $\beta$  from Vinken et al. [2014] calculated for 2006 and include a discussion (in section 3.5 + Supplement 4) on variability in  $\beta$  from year to year, and how this could have influenced the top-down NOx emission estimates.

To evaluate possible changes in chemical regime between 2019 and 2020, we studied the EAC4 re-analysis of atmospheric composition (Inness et al., 2021). NO2 and O3 monthly mean columns show only modest year-to-year variability between 2006 and 2019, as shown in Figure 1 below. This suggests modest variability in the chemical regimes (and therefore  $\beta$ ) between 2006 and 2019.